

# Global transpiration data from sap flow measurements: the SAPFLUXNET database

Rafael Poyatos[1,2], Víctor Granda[1,3], Víctor Flo[1], Mark A. Adams[4,5], Balázs Adorján[6], David Aguadé[1], Marcos P.M. Aidar[7], Scott Allen[8], M. Susana Alvarado-Barrientos[9], Kristina J. Anderson-Teixeira[10,11], Luiza Maria Aparecido[12,13], M. Altaf Arain[14], Ismael Aranda[15], Heidi Asbjornsen[16], Robert Baxter[17], Eric Beamesderfer[18,19], Z. Carter Berry[20], Daniel Berveiller[21], Bethany Blakely[22], Johnny Boggs[23], Gil Bohrer[24], Paul V. Bolstad[25], Damien Bonal[26], Rosvel Bracho[27], Patricia Brito[28], Jason Brodeur[29], Fernando Casanoves[30], Jérôme Chave[31], Hui Chen[32], Cesar Cisneros[33,34], Kenneth Clark[35], Edoardo Cremonese[36], Jorge S. David[37], Teresa S. David[37,38], Nicolas Delpierre[39,40], Ankur R. Desai[41], Frederic C. Do[42], Michal Dohnal[43], Jean-Christophe Domec[44,45], Sebinasi Dzikiti[46], Colin Edgar[47], Rebekka Eichstaedt[48,**], Tarek S. El-Madany[49], Jan Elbers[50], Cleiton B. Eller[51], Eugénie S. Euskirchen[47], Brent Ewers[52], Patrick Fonti[53], Alicia Forner[54], David I. Forrester[53], Helber C. Freitas[55,56], Marta Galvagno[36], Omar Garcia-Tejera[57], Chandra Prasad Ghimire[58,34], Teresa E. Gimeno[59,60], John Grace[61], André Granier[62], Anne Griebel[63,64], Yan Guangyu[32], Mark B. Gush[65], Paul Hanson[66], Niles J. Hasselquist[67], Ingo Heinrich[68], Virginia Hernandez-Santana[69], Valentine Herrmann[70], Teemu Hölttä[71], Friso Holwerda[72], Dang Hongzhong[73], James Irvine[61], Supat Isarangkool Na Ayutthaya[74], Paul G. Jarvis[61,*], Hubert Jochheim[75], Carlos A. Joly[76,77], Julia Kaplick[78,79], Hyun Seok Kim[80,81,82], Leif Klemedtsson[83], Heather Kropp[84,85], Fredrik Lagergren[86], Patrick Lane[87], Petra Lang[88], Andrei Lapenas[89], Víctor Lechuga[90], Minsu Lee[80], Christoph Leuschner[91], Jean-Marc Limousin[92], Juan Carlos Linares[93], Maj-Lena Linderson[86], Anders Lindroth[86], Pilar Llorens[94], Álvaro López-Bernal[95], Michael M. Loranty[96], Dietmar Lüttschwager[75], Cate Macinnis-Ng[79], Isabelle Maréchaux[97], Timothy A. Martin[98], Ashley Matheny[99], Nate McDowell[100], Sean McMahon[101], Patrick Meir[102,61], Ilona Mészáros[6], Mirco Migliavacca[49], Patrick Mitchell[103], Meelis Mölder[104], Leonardo Montagnani[105,106], Georgianne W. Moore[107], Ryogo Nakada[108], Furong Niu[109,110], Rachael H. Nolan[63], Richard Norby[111], Kimberly Novick[112], Walter Oberhuber[113], Nikolaus Obojes[114], A. Christopher Oishi[115], Rafael S. Oliveira[51], Ram Oren[116,117], Jean-Marc Ourcival[92], Teemu Paljakka[118], Oscar Perez-Priego[119,49], Pablo L. Peri[120,121,122], Richard L. Peters[123,53], Sebastian Pfautsch[124], William T. Pockman[125], Yakir Preisler[126], Katherine Rascher[127], George Robinson[128], Humberto Rocha[129], Alain Rocheteau[42], Alexander Röll[110], Bruno Rosado[130], Lucy Rowland[131], Alexey V. Rubtsov[132], Santiago Sabaté[1,133], Yann Salmon[134,135], Roberto L. Salomón[136,137], Elisenda Sánchez-Costa[138], Karina V.R. Schäfer[139], Bernhard Schuldt[140], Alexandr Shashkin[141], Clément Stahl[142], Marko Stojanović[143], Juan Carlos Suárez[144,145], Ge Sun[23], Justyna Szatniewska[143], Fyodor Tatarinov[126], Miroslav Tesař[146], Frank M. Thomas[147], Pantana Tor-ngern[148,149], Josef Urban[150,151], Fernando Valladares[152,153], Christiaan van der Tol[154], Ilja van Meerveld[155], Andrej Varlagin[156], Holm Voigt[157], Jeffrey Warren[158], Christiane Werner[159], Willy Werner[160], Gerhard Wieser[161], Lisa Wingate[162], Stan Wullschleger[163], Koong Yi[164,165], Roman Zweifel[166], Kathy Steppe[167], Maurizio Mencuccini[1,168], Jordi Martínez-Vilalta[1,2]

[1]CREAF, E08193 Bellaterra (Cerdanyola del Vallès), Catalonia, Spain.
[2]Universitat Autònoma de Barcelona, E08193 Bellaterra, (Cerdanyola del Vallès), Catalonia, Spain.





[3]Joint Research Unit CREAF-CTFC, Bellaterra, Catalonia, Spain.
[4]Faculty of Science Engineering and Technology, Swinburne University of Technology, Hawthorn, Vic 3122 AUSTRALIA.
[5]School of Life and Environmental Sciences, University of Sydney, Camperdown, New South Wales, Australia.
[6]Department of Botany, University of Debrecen, Faculty of Science and Technology, Egyetem tér 1, 4032 Debrecen, Hungary.
[7]Plant Physiology and Biochemistry Reserach Nucleus, Institute fo Botany, São Paulo, Brazil.
[8]Department of Natural Resources and Environmental Science. University of Nevada, Reno, USA.
[9]Red Ecología Funcional, Instituto de Ecología A.C., Xalapa, México.
[10]Center for Tropical Forest Science-Forest Global Earth Observatory; Smithsonian Tropical Research Institute; Panama, Republic of Panama.
[11]Conservation Ecology Center, Smithsonian Conservation Biology Institute, Front Royal, VA, USA.
[12]Department of Ecosystem Science and Management, Texas A&M University, College Station, Texas, USA.
[13]School of Earth and Space Exploration, Arizona State University, Tempe, Arizona, USA.
[14]School of Earth, Environment & Society and McMaster Centre for Climate Change, McMaster University, Hamilton, Ontario, Canada.
[15]National Institute for Agricultural and Food Research and Technology (INIA), Forest Reseach Centre (CIFOR), Department of Forest Ecology and Genetics, Avda. A Coruña km 7.5, E-28040 Madrid, Spain.
[16]Department of Natural Resources and the Environment, University of New Hampshire, Durham, NH USA.
[17]Department of Biosciences, University of Durham, Durham, UK.
[18]School of Geography and Earth Sciences and McMaster Centre for Climate Change, McMaster University, Hamilton, Ontario, Canada.
[19]School of Informatics, Computing & Cyber Systems, Northern Arizona University, Flagstaff, Arizona, USA.
[20]Schmid College of Science and Technology, Chapman University, Orange, CA 92866, USA.
[21]Université Paris-Saclay, CNRS, AgroParisTech, Ecologie Systématique et Evolution, 91405, Orsay, France.
[22]University of Illinois at Urbana-Champaign, Urbana-Champaign, IL, USA.
[23]Eastern Forest Environmental Threat Assessment Center, Southern Research Station, USDA Forest Service, Research Triangle Park, NC 27709, USA.
[24]Department of Civil, Environmental and Geodetic Engineering, Ohio State University, 405 Hitchcock Hall, 2070 Neil Avenue, Columbus, OH 43210, USA.
[25]Department of Forest Resources, University of Minnesota, Saint Paul, MN, USA.
[26]Université de Lorraine, INRAE, AgroParisTech, 54000 Nancy, France.
[27]School of Forest Resources and Conservation, University of Florida, Gainesville, FL 32611, USA.
[28]Department of Botany, Ecology and Plant Physiology, University of La Laguna (ULL), Apdo. 456, 38200 La Laguna, Tenerife, Spain.
[29]McMaster University Library, McMaster University, Hamilton, Ontario, Canada.
[30]CATIE-Centro Agronómico Tropical de Investigación y Enseñanza, Costa Rica.
[31]Laboratoire Evolution and Diversité Biologique, CNRS, UPS, IRD, Bâtiment 4R1 Université Paul Sabatier, 118 route de
Narbonne, 31062 Toulouse cedex 4, France.
[32]Key Laboratory of the Ministry of Education for Coastal and Wetland Ecosystems, School of Life Sciences, Xiamen University, Xiamen, Fujian 361005, China.
[33]Carrera de Ingeniería Ambiental, Facultad de Ingeniería, Universidad Nacional de Chimborazo, EC060108, Riobamba, Ecuador.
[34]Faculty of Geo-information and Earth Observation (ITC), University of Twente, Enschede, Hengelosestraat 99, 7514 AE Enschede, The Netherlands.
[35]USDA Forest Service, Northern Research Station, Silas Little Experimental Forest, New Lisbon, New Jersey 08064 USA.
[36]Climate Change Unit, Environmental Protection Agency of Aosta Valley, Saint Christophe, 11020, Italy.
[37]Centro de Estudos Florestais, Instituto Superior de Agronomia, Universidade de Lisboa, Tapada da Ajuda, 1349-017
Lisboa, Portugal.
[38]Instituto Nacional de Investigação Agrária e Veterinária I.P., Quinta do Marquês, Av. da República, 2780-159 Oeiras, Portugal.



[39]Institut Universitaire de France (IUF), 75231 PARIS, France.

[40]Université Paris-Saclay, CNRS, AgroParisTech, Ecologie Systématique et Evolution, 91405, Orsay, France..

[41]Dept of Atmospheric and Oceanic Sciences, University of Wisconsin-Madison, 1225 W Dayton St, Madison, WI 53706 USA.

[42]Eco&Sols, Univ Montpellier, CIRAD, INRAE, Institut Agro, IRD, 34060 Montpellier, France.

[43]Czech Technical University in Prague, Faculty of Civil Engineering, Thakurova 7, 16629, Prague, Czech Republic.

[44]Bordeaux Sciences Agro, UMR 1391 INRA-BSA, France.

[45]Nicholas School of the Environment, Duke University, USA.

[46]Department of Horticultural Science, University of Stellenbosch, South Africa.

[47]University of Alaska Fairbanks, Institute of Arctic Biology, Fairbanks, AK 99775, USA.

[48]Faculty of Regional and Environmental Sciences – Geobotany, University of Trier, Behringstraße 21, Trier 54296, Germany.

[49]Max Planck Institute for Biogeochemistry, Hans-Knöll-Str. 10, Jena, Germany.

[50]Wageningen University and Research, Water Systems and Global Change Group, P.O. Box 47, 6700AA Wageningen, The Netherlands.

[51]Department of Plant Biology, University of Campinas, Campinas 13083-862, Brazil.

[52]Department of Botany, University of Wyoming, Laramie, WY, USA.

[53]Swiss Federal Institute for Forest, Snow and Landscape Research WSL, Zuercherstrasse 111, 8903 Birmensdorf, Switzerland.

[54]Laboratorio Internacional de Cambio Global (LINCGlobal), Departamento de Biogeografía y Cambio Global, Museo Nacional de Ciencias Naturales, MNCN, CSIC, C/Serrano 115 dpdo, E-28006 Madrid, Spain.

[55]São Paulo State University (Unesp), School of Sciences, Bauru, Brazil.

[56]University of São Paulo, Institute of Astronomy, Geophysics and Atmospheric Sciences, São Paulo, Brazil.

[57]Efficient Use of Water Program, Institut de Recerca i Tecnologia Agroalimentàries (IRTA), Parc de Gardeny, Edifici Fruitcentre, 25003 Lleida, Spain.

[58]AgResearch, Lincoln Research Centre, Private bag 4749, Christchurch 8140, New Zealand.

[59]Basque Centre for Climate Change (BC3), 48940 Leioa, Spain.

[60]Basque Foundation for Science, 48008, Bilbao, Spain.

[61]School of Geosciences, University of Edinburgh, Edinburgh UK.

[62]NRAE, UMR SILVA 1434, Champenoux, 54280, France.

[63]Hawkesbury Institute for the Environment, Western Sydney University, Sydney, NSW, Australia.

[64]School of Ecosystem and Forest Sciences, The University of Melbourne, 500 Yarra Boulevard, Richmond, VIC 3121, 120 Australia.

[65]Science & Collections Division, Royal Horticultural Society, Wisley, Woking, Surrey, GU23 6QB, United Kingdom.

[66]Environmental Sciences Division, Oak Ridge National Laboartory, Oak Ridge, Tennessee 37831 USA.

[67]Department of Forest Ecology and Management, Swedish University of Agricultural Sciences, Umeå, Sweden.

[68]Section Climate Dynamics and Landscape Evolution, Helmholtz Centre Potsdam, GFZ German Research Centre for 125 Geosciences, 14473 Potsdam, Germany.

[69]Irrigation and Crop Ecophysiology Group, Instituto de Recursos Naturales y Agrobiología de Sevilla (IRNAS, CSIC). Avenida Reina Mercedes, n.º 10, 41012-Sevilla, Spain.

[70]Conservation Ecology Center; Smithsonian Conservation Biology Institute; Front Royal, VA, USA.

[71]Institute for Atmospheric and Earth System Research/Forest Sciences, Faculty of Agriculture and Forestry, University of 130 Helsinki, Finland.

[72]Centro de Ciencias de la Atmósfera, Universidad Nacional Autónoma de México, Mexico City, Mexico.

[73]Institute of Desertification Studies, Chinese Academy of Forestry, Beijing 100091, China.

[74]Department of Horticulture, Faculty of Agriculture, Khon Kaen University, Khon Kaen, Thailand.

[75]Leibniz Centre for Agricultural Landscape Research (ZALF), Eberswalder Str. 84, 15374 Müncheberg, Germany.

[76]Brazilian Platform of Biodiversity and Ecosystem Services/BPBES, Brazil.

[77]Departamento de Biologia Vegetal, Instituto de Biologia, Universidade Estadual de Campinas, Campinas, São Paulo, Brazil.



[78]Head Office of Forest Protection, Brandenburg State Forestry Center of Excellence, 16225 Eberswalde, Germany.
[79]School of Biological Sciences, University of Auckland, Auckland, New Zealand.
[80]Department of Forest Sciences, Seoul National Univeristy, Seoul, Rep of Korea.
[81]National Center for Agro Meteorology, Seoul, Rep of Korea.
[82]Research Institute for Agriculture and Life Sciences, Seoul National University, Seoul, Rep of Korea.
[83]Department of Earth Sciences, Gothenburg Univ., Guldhedsgatan 5A, PO Box 460, SE 405 30 Gothenburg, Sweden.
[84]Environmental Studies, Hamilton College, Clinton, NY, USA.
[85]Geography Department, Colgate University, Hamilton, NY, USA.
[86]Department of Physical Geography and Ecosystem Science, Lund University, Lund, Sweden.
[87]School of Ecosystem and Forest Sciences, The University of Melbourne, Parkville, Victoria, 3010 Australia.
[88]Landeshauptstadt München, Referat für Gesundheit und Umwelt, Nachhaltige Entwicklung, Umweltplanung, SG Ressourcenschutz, 80335 München, Germany.
[89]Department of Geography and Planning, University at Albany, NY, USA.
[90]Department of Animal Biology, Vegetal Biology and Ecology. University of Jaén, Spain.
[91]Plant Ecology, University of Goettingen, 37073 Goettingen, Germany.
[92]CEFE, Univ Montpellier, CNRS, EPHE, IRD, Univ Paul Valéry Montpellier 3, Montpellier, France.
[93]Department of Physical, Chemical and Natural Systems,University Pablo de Olavide, 41013 Seville, Spain.
[94]Surface Hydrology and Erosion group, Institute of Environmental Assessment and Water Research, CSIC, Barcelona, Spain.
[95]Departamento de Agronomía, Universidad de Córdoba, 14071 Córdoba, Spain.
[96]Department of Geography, Colgate University, Hamilton, NY, USA.
[97]AMAP, Univ Montpellier, CIRAD, CNRS, INRAE, IRD, 34000 Montpellier, France.
[98]University of Florida, School of Forest Resources and Conservation, 136 Newins-Ziegler Hall, Gainesville, FL 32611, USA.
[99]Department of Geological Sciences, Jackson School of Geosciences, University of Texas at Austin, Austin, Texas, USA.
[100]Pacific Northwest National Laboratory, Richland, WA, USA.
[101]Center for Tropical Forest Science-Forest Global Earth Observatory, Smithsonian Environmental Research Center,
Edgewater, MD, 21307 USA.
[102]Research School of Biology, Australian National University, ACT 2601 Australia.
[103]CSIRO Agriculture and Food, Sandy Bay, Tasmania 7005, Australia.
[104]Dept of Physical Geography and Ecosystem Science, University of Lund, Sweden.
[105]Faculty of Science and Technilogy, Free University of Bolzano, Piazza Università 5, Bolzano, Italy.
[106]Forest Services, Autonomous Province of Bolzano, Bolzano, Italy.
[107]Department of Ecology and Conservation Biology, Texas A&M University, College Station, Texas, USA.
[108]Hokkaido Regional Breeding Office, Forest Tree Breeding Center, Forestry and Forest Products Research Institute, Ebetsu, Hokkaido, Japan.
[109]School of Natural Resources and the Environment, University of Arizona, Tucson, AZ 85721, USA.
[110]Tropical Silviculture and Forest Ecology, University of Goettingen, Büsgenweg 1, 37077 Göttingen, Germany.
[111]Department of Ecology & Evolutionary Biology, University of Tennessee, Knoxville, TN USA.
[112]O'Neill School of Public and Environmental Affairs, Indiana University-Bloomington, Bloomington, IN, USA.
[113]University of Innsbruck, Department of Botany, Sternwartestrasse 15, 6020 Innsbruck, Austria.
[114]EURAC Research, Institute for Alpine Environment, Viale Druso 1, Bolzano, Italy.
[115]USDA Forest Service, Southern Research Station, Coweeta Hydrologic Laboratory, Otto, NC, USA.
[116]Department of Forest Sciences, University of Helsinki, P.O. Box 27, Helsinki FI-00014, Finland.
[117]Division of Environmental Science & Policy, Nicholas School of the Environment, and Department of Civil & Environmental Engineering, Pratt School of Engineering, Duke University, Durham, North Carolina, USA.
[118]Institute for Atmospheric and Earth System Research / Forest Sciences, Faculty of Agriculture and Forestry, University of
Helsinki, Helsinki, Finland.
[119]Biological sciences department, Macquarie University, Sydney, NSW, Australia.
[120]National Institute of Agricultural Technology (INTA), CC 332, CP 9400, Río Gallegos, Santa Cruz, Argentina.



[121]National Scientific and Technical Research Council of Argentina (CONICET).

[122]National University of Southern Patagonia (UNPA).

[123]Laboratory of Plant Ecology, Faculty of Bioscience Engineering, Ghent University, Coupure links 653, B-9000 Ghent, Belgium.

[124]Urban Studies, School of Social Sciences, Western Sydney University, Locked Bag 1797, NSW 2751, Australia.

[125]Department of Biology, University of New Mexico, Albuquerque, NM, USA.

[126]The Earth and Planetary Science Department, Weizmann Institute of Science, Rehovot, Israel.

[127]University of Cologne, Faculty of Medicine and University Hospital Cologne, Cologne, Germany.

[128]Department of Biological Science, University at Albany, NY, USA.

[129]Laboratorio de Clima e Biosfera, Instituto de Astronomia Geofisica e Ciencias Atmosfericas, Universidade de Sao Paulo, Brazil.

[130]Department of Ecology, IBRAG, Universidade do Estado do Rio de Janeiro (UERJ). R. São Francisco Xavier, 524, PHLC,
Sala 220. CEP 20550900, Maracanã, Rio de Janeiro, RJ, Brazil.

[131]College of life and environmental science, University of Exeter, Laver Building, North Park Road, EX4 4QE, United Kingdom.

[132]Laboratory for integral studies of forest dynamics of Eurasia, Siberian Federal University, Krasnoyarsk, Russia.

[133]Department of Evolutionary Biology, Ecology, and Environmental Sciences, University of Barcelona (UB), Barcelona
08028, Spain.

[134]Institute for Atmospheric and Earth System Research (INAR)/Forest, University of Helsinki, 00014 University of Helsinki, Finland.

[135]Institute for Atmospheric and Earth System Research (INAR)/Physics, University of Helsinki, 00014 University of Helsinki, Finland.

[136]Forest Genetics and Ecophysiology Research Group, Universidad Politécnica de Madrid, Ciudad Universitaria s/n, 28040 Madrid, Spain,.

[137]Laboratory of Plant Ecology, Department of Plants and Crops, Faculty of Bioscience Engineering, Ghent University, Coupure Links 653, 9000 Ghent, Belgium.

[138]IRTA, Institute of Agrifood Research and Technology, Torre Marimon, E-08140 Caldes de Montbui, Barcelona, Spain.

[139]Earth and Environmental Science Department, Rutgers University Newark, 195 University Av, Newark, NJ 07102, USA.

[140]University of Würzburg, Julius-von-Sachs-Institute for Biological Sciences, Chair of Ecophysiology and Vegetation Ecology, Julius-von-Sachs-Platz 3, 97082 Würzburg, Germany.

[141]Sukachev Institute of Forest of the Siberian Branch of the RAS, Krasnoyarsk, Russian Federation.

[142]UMR EcoFoG, CNRS, CIRAD, INRAE, AgroParisTech, Université des Antilles, Université de Guyane, 97310 Kourou,
France.

[143]Global Change Research Institute of the Czech Academy of Sciences, Bělidla 4a, 603 00, Brno, Czech Republic.

[144]Centro de Investigaciones Amazónicas CIMAZ Macagual César Augusto Estrada González, Grupo de Investigaciones Agroecosistemas y Conservación en Bosques Amazónicos-GAIA, Florencia, Caquetá, Colombia.

[145]Universidad de la Amazonia, Programa de Ingeniería Agroecológica, Facultad de Ingeniería, Florencia, Caquetá,
Colombia.

[146]Institute of Hydrodynamics, Academy of Sciences of the Czech Republic, Prague, Czech Republic.

[147]Trier University, Faculty of Regional and Environmental Sciences, Geobotany, Behringstr. 21, D-54296, Trier, Germany.

[148]Department of Environmental Science, Faculty of Science, Chulalongkorn University, Bangkok 10330 Thailand.

[149]Environment, Health and Social Data Analytics Research Group, Chulalongkorn University, Bangkok 10330 Thailand.

[150]Department of Forest Botany, Dendrology and Geobiocenology, Faculty of Forestry and Wood Technology, Mendel University in Brno, Zemedelska 3, 61300 Brno, Czech Republic.

[151]Laboratory for Complex Studies of Forest Dynamics in Eurasia, Siberian Federal University, Akademgorodok 50A-K2, Krasnoyarsk, Russia.

[152]Departamento de Biología y Geología, Escuela Superior de Ciencias Experimentales y Tecnológicas, Universidad Rey
Juan Carlos, C/Tulipán s/n, E-28933 Móstoles, Spain..

[153]Laboratorio Internacional de Cambio Global (LINCGlobal), Departamento de Biogeografía y Cambio Global, Museo Nacional de Ciencias Naturales, MNCN, CSIC, C/Serrano 115 dpdo, E-28006 Madrid, Spain..





[154]University of Twente, Faculty ITC, P.O. Box 217, 7500 AE Enschede, The Netherlands.
[155]Department of Geography, Hydrology and Climate, University of Zurich, Winterthurerstrasse 190, 8057 Zurich,
Switzerland.
[156]A.N. Severtsov Institute of Ecology and Evolution, Russian Academy of Sciences, 119071, Leninsky pr.33, Moscow,
Russia.
[157]ZEF Center for Development Research, University of Bonn, Genscherallee 3, 53113 Bonn, Germany.
[158]Environmental Sciences Division, Oak Ridge National Laboratory, Oak Ridge, TN 37831 USA.
[159]Ecosystem Physiology, University of Freiburg, 79098Freiburg, Germany.
[160]Geobotany Department, University of Trier, 54286, Trier, Germany.
[161]Division of Alpine Timberline Ecophysiology, Federal Research and Training Centre for Forests, Natural Hazards and
Landscape (BFW), Rennerg 1, 6020 Innsbruck, Austria.
[162]INRAE, UMR ISPA 1391, Villenave D'Ornon, 33140, France.
[163]Environmental Sciences Division, Oak Ridge National Laboratory, Oak Ridge, TN 37831, USA.
[164]Department of Environmental Sciences, University of Virginia, Charlottesville, VA 22904, USA.
[165]O'Neill School of Public and Environmental Affairs, Indiana University Bloomington, Bloomington, IN 47405, USA.
[166]Swiss Federal Institute for Forest, Snow and Landscape Research WSL, Birmensdorf, Switzerland.
[167]Laboratory of Plant Ecology, Faculty of Bioscience Engineering, Ghent University, B-9000 Ghent, Belgium.
[168]ICREA, Barcelona, Catalonia, Spain.
[*] Deceased
[**]Previously published under the name Rebekka Boegelein

*Correspondence to*: Rafael Poyatos (r.poyatos@creaf.uab.cat)

**Abstract.** Plant transpiration links physiological responses of vegetation to water supply and demand with hydrological,
energy and carbon budgets at the land-atmosphere interface. However, despite being the main land evaporative flux at the
global scale, transpiration and its response to environmental drivers are currently not well constrained by observations. Here
we introduce the first global compilation of whole-plant transpiration data from sap flow measurements (SAPFLUXNET,
https://sapfluxnet.creaf.cat/). We harmonised and quality-controlled individual datasets supplied by contributors worldwide
in a semi-automatic data workflow implemented in the R programming language. Datasets include sub-daily time series of
sap flow and hydrometeorological drivers for one or more growing seasons, as well as metadata on the stand characteristics,
plant attributes and technical details of the measurements. SAPFLUXNET contains 202 globally distributed datasets with
sap flow time series for 2714 plants, mostly trees, of 174 species. SAPFLUXNET has a broad bioclimatic coverage, with
woodland/shrubland and temperate forest biomes especially well-represented (80% of the datasets). The measurements cover
a wide variety of stand structural characteristics and plant sizes. The datasets encompass the period between 1995 and 2018,
with 50% of the datasets being at least 3 years long. Accompanying radiation and vapour pressure deficit data are available
for most of the datasets, while on-site soil water content is available for 56% of the datasets. Many datasets contain data for
species that make up 90% or more of the total stand basal area, allowing the estimation of stand transpiration in diverse
ecological settings. SAPFLUXNET adds to existing plant trait datasets, ecosystem flux networks and remote sensing
products to help increase our understanding of plant water use, plant responses to drought and ecohydrological processes.
SAPFLUXNET version 0.1.5 is freely available from the Zenodo repository (https://doi.org/10.5281/zenodo.3971689,





Poyatos et al. 2020a). The 'sapfluxnetr' R package, designed to access, visualise and process SAPFLUXNET data is available from CRAN.

## 1 Introduction

Terrestrial vegetation transpires *ca.* 45000 km$^3$ of water per year (Schlesinger and Jasechko, 2014; Wang-Erlandsson et al., 2014; Wei et al., 2017), a flux that represents 40% of global land precipitation, 70% of total land evapotranspiration (Oki and Kanae, 2006), and is comparable in magnitude to global annual river discharge (Rodell et al., 2015). For most terrestrial plants, transpiration is an inevitable water loss to the atmosphere because they need to open stomata to allow $CO_2$ diffusion into the leaves for photosynthesis. Latent heat from transpiration represents 30–40% of surface net radiation globally (Schlesinger and Jasechko, 2014; Wild et al., 2015). Transpiration is therefore a key process coupling land-atmosphere exchange of water, carbon and energy, determining several vegetation-atmosphere feedbacks, such as land evaporative cooling or moisture recycling. Regulation of transpiration in response to fluctuating water availability and/or evaporative demand is a key component of plant functioning and one of the main determinants of a plant's response to drought (Martin-StPaul et al., 2017; Whitehead, 1998). Despite its relevance for earth functioning, transpiration and its spatiotemporal dynamics are poorly constrained by available observations (Schlesinger and Jasechko, 2014) and not well represented in models (Fatichi et al., 2016; Mencuccini et al., 2019). An improved understanding on how plants regulate transpiration is thus needed to better predict future trajectories of land evaporative fluxes and vegetation functioning under increased drought conditions driven by global change.

Conceptually, transpiration can be quantified at different organisational scales: leaves, branches and whole plants, ecosystems and watersheds. In practice, transpiration is relatively easy to isolate from the bulk evaporative flux, evapotranspiration, only from the leaf to the plant levels. In terrestrial ecosystems, evapotranspiration includes evaporation from the soil and from water-covered surfaces, including plants. Transpiration measurements on individual leaves or branches with gas exchange systems are difficult to upscale to the plant level (Jarvis, 1995). Likewise, transpiration measurements using whole-plant chambers  (e.g. Pérez-Priego et al., 2010) or gravimetric methods (e.g. weighing lysimeters) in the field are still challenging. At the ecosystem scale and beyond, evapotranspiration is generally determined using micrometeorological methods, catchment water budgets or remote sensing approaches (Shuttleworth, 2007; Wang and Dickinson, 2012). In some cases, isotopic methods and different algorithms applied to measured ecosystem fluxes can provide an estimation of transpiration at the ecosystem scale (Kool et al., 2014; Stoy et al., 2019).

Transpiration drives water transport from roots to leaves in the form of sap flow through the plant's xylem pathway (Tyree and Zimmermann, 2002), and this sap flow affects heat transport in the xylem. Taking advantage of this, thermometric sap flow methods were first developed in the 1930s (Huber, 1932) and further refined over the following decades (Čermák et al.,



1973; Marshall, 1958) to provide operational measurements of plant water use. These methods have become widely used in plant ecophysiology, agronomy and hydrology (Poyatos et al., 2016), especially after the development of simple, easily

replicable methods (e.g. Granier, 1985, 1987). Whole-plant measurements of water use using thermometric sap flow methods provide estimates of water flow through plants from sub-daily to interannual timescales, and have been mostly applied in woody plants (but see Baker and Van Bavel (1987) for measurements on herbaceous species). Xylem sap flow is measured semi-invasively (Brodersen et al., 2019) and can be upscaled to the whole plant, obtaining a near-continuous quantification of plant water use. Multiple sap flow sensors can be deployed, in almost any terrestrial ecosystem, to

determine the magnitude and temporal dynamics of transpiration across species, environmental conditions or experimental treatments. All sap flow methods are subject to methodological and scaling issues, which may affect the quantification of absolute water use in some circumstances (Čermák et al., 2004; Köstner et al., 1998; Smith and Allen, 1996; Vandegehuchte and Steppe, 2013). Nevertheless, all methods are suitable for the assessment of the temporal dynamics of transpiration and of its responses to environmental changes or to experimental treatments (Flo et al., 2019).


The generalised application of sap flow methods in ecological and hydrological research in the last 30 years has thus generated a large volume of data, with an enormous potential to advance our understanding of the spatiotemporal patterns and the ecological drivers of plant transpiration and its regulation (Poyatos et al., 2016). However, this large volume of data needs to be compiled and harmonised to enable global syntheses and comparative studies across species and regions. Across-

species data syntheses using sap flow data have mostly focused on maximum values extracted from publications (Kallarackal et al., 2013; Manzoni et al., 2013; Wullschleger et al., 1998). Multi-site syntheses have focused on the environmental sensitivity of sap flow, using site means of plant-level sap flow or sap flow-derived stand transpiration (Poyatos et al., 2007; Tor-ngern et al., 2017). Since data sharing is only incipient in plant ecophysiology, sap flow datasets have not been traditionally available in open data repositories. Open data practices are now being implemented in databases, which fosters

collaboration across monitoring networks in research areas relevant to plant functional ecology (Falster et al., 2015; Gallagher et al., 2020; Kattge et al., 2020) and ecosystem ecology (Bond-Lamberty and Thomson, 2010). The success of the data sharing and data re-use policies within the FLUXNET global network of ecosystem level fluxes has shown how these practices can contribute to scientific progress (Bond-Lamberty, 2018).

Here we introduce SAPFLUXNET, the first global database of sap flow measurements built from individual community-contributed datasets. We implemented this compilation in a data structure designed to accommodate time series of sap flow and the main hydrometeorological drivers of transpiration, together with metadata documenting different aspects of each dataset. We harmonised all datasets and performed basic semi-automated quality assurance and quality control procedures. We also created a software package that provides access to the database, allows easy visualisation of the datasets and

performs basic temporal aggregations. We present the ecological and geographic coverage of SAPFLUXNET version 0.1.5,





(Poyatos et al., 2020a) followed by a discussion of potential applications of the database, its limitations and a perspective of future developments.

## 2 The SAPFLUXNET data workflow

### 2.1 An overview of sap flow measurements

The main characteristics of sap flow methods have been reviewed elsewhere (Čermák et al., 2004; Smith and Allen, 1996; Swanson, 1994; Vandegehuchte and Steppe, 2013). Given the already broad scope of the paper, here we only provide a brief methodological overview, without delving into the details of the individual methods. Sap flow sensors track the fate of heat applied to the plant's conducting tissue, or sapwood, using temperature sensors (thermocouples or thermistors), usually deployed in the plant's main stem. Both heating and temperature sensing can be done either internally, by inserting needle-

like probes containing electrical resistors (or electrodes for some methods) and temperature sensors into the sapwood, or externally; these latter systems being especially designed for small stems. Depending on how the heat is applied and the principles underlying sap flow calculations, sap flow sensors can be classified into three major groups: heat dissipation methods, heat pulse methods and heat balance methods (Flo et al., 2019). Heat dissipation and heat pulse methods estimate sap flow per unit sapwood area and they have been called 'sap flux density methods' (Vandegehuchte and Steppe, 2013);

heat balance methods directly yield sap flow for the entire stem or for a sapwood section. Heat dissipation methods include the constant heat dissipation (HD; Granier 1985, 1987), the transient (or cyclic) heat dissipation (CHD; Do and Rocheteau, 2002) and the heat deformation (HFD; Nadezhdina 2018) methods. Heat pulse methods include the compensation heat pulse (CHP; Swanson and Whitfield, 1981), heat ratio (HR; Burgess et al. 2001), T-max (HPTM; Cohen et al. 1981) and Sapflow+ (Vandegehuchte and Steppe, 2012) methods. Heat balance methods include the trunk sector heat balance (TSHB; Čermák et

al. 1973) and the stem heat balance (SHB; Sakuratani, 1981) methods. The suitability of a certain method in a given application largely depends on plant size and the flow range of interest (Flo et al., 2019), but HD and CHP are the most widely used (Flo et al., 2019; Peters et al., 2018; Poyatos et al., 2016). Apart from these different methodologies, within each sap flow method variants exist in sensor design and in data processing approaches, resulting in relatively high levels of methodological uncertainty comparable to those in other areas of plant ecophysiology.


The output from sap flow sensors is automatically recorded by dataloggers, at hourly or even higher temporal resolution. This output relates to heat transport in the stem and needs to be converted to meaningful quantities of water transport, such as sap flow per plant or per unit sapwood area. How this conversion is achieved varies greatly across methods, with some relying on empirical calibrations and others being more physically-based and requiring the estimation of wood thermal

properties and other parameters (Čermák et al., 2004; Smith and Allen, 1996; Vandegehuchte and Steppe, 2013). Depending on the method and the specific sensor design, sap flow measurements can be representative of single points, linear segments along the sapwood, sapwood area sections or entire stems. Except for stem heat balance methods, these measurements need





to be spatially integrated to account for radial (Berdanier et al., 2016; Cohen et al., 2008; Nadezhdina et al., 2002; Phillips et al., 1996) and azimuthal (Cohen et al., 2008; Lu et al., 2000; Oren et al., 1999a) variation of sap flow within the stem to obtain an estimate of whole-plant water use (Čermák et al., 2004). At a minimum, an estimate of sapwood area is needed to upscale the measurements to whole-plant sap flow rates. Sap flow rates can thus be expressed per individual (i.e. plant or tree), per unit sapwood area (normalising by water-conducting area), and per unit leaf area (normalising by transpiring area).

Here we will use the term 'sap flow' when referring, in general, to the rate at which water moves through the sapwood of a plant and, more specifically, when we refer to sap flow per plant (i.e. water volume per unit time, Edwards et al., 1996). We acknowledge that the term 'sap flux' has also been proposed for this quantity (Lemeur et al., 2009), but more generally, 'sap flux density' (e.g. Vandegehuchte and Steppe, 2013) or just 'sap flux' are used to refer to 'sap flow per unit sapwood area'. Since here we include methods natively measuring sap flow per plant or per sapwood area, throughout this paper we will use the more general term 'sap flow', and, when necessary, we will indicate explicitly the reference area used: 'sap flow per (unit) sapwood area', 'sap flow per (unit) leaf area' or 'sap flow per (unit) ground area'.

## 2.2 Data compilation

SAPFLUXNET was conceived as a compilation of published and unpublished sap flow datasets (Appendix Table A1) and thus the ultimate success of the initiative critically depended on the contribution of datasets by the sap flow community. An expression of interest showed that a critical mass of datasets with a wide geographic distribution could potentially be contributed and the results of this survey were used to raise the interest of the sap flow community (Poyatos et al., 2016). The data contribution stage was open between July 2016 and December 2017 although a few additional datasets were updated during the data quality control process and contain more recent data.

All contributed datasets had to meet some minimum criteria before they were accepted, both in terms of content and format. We required that all datasets contained sub-daily, processed sap flow data, representative of whole-plant water use under different hydrometeorological conditions. This meant that both the processing from raw temperature data to sap flow quantities and the scaling from single-point measurements to whole-plant data had been performed by the data contributor responsible for each dataset. Time-series of sap flow data and hydrometeorological drivers were required to be representative of one growing-season, setting, as broad reference, a minimum duration of 3 months. Sap flow could be either expressed as total flow rate per plant or per unit sapwood area. Contributors also needed to provide metadata on relevant ecological information of the site, stand, species and measured plants as well as on basic technical details of the sap flow and hydrometeorological time-series. Datasets had to be formatted using a documented spreadsheet template (cf. 'sapfluxnet_metadata_template.xlsx' in the Supplement) and uploaded to a dedicated server at CREAF, Spain, using an online form.





## 2.3 Data harmonisation and quality control: QC1

Once datasets were received, they were stored and entered a process of data harmonisation and quality control (Fig. 1, Supplement Fig. S1). This process combined automatic data checks with human supervision, and the entire workflow was governed by functions and scripts in the R language (R Core Team, 2019), including other related tools, such as R

markdown documents and Shiny applications. All R code involved in this QC process was implemented in the sapfluxnetQC1 package (Granda et al., 2016). To aid in the detection of potential data issues throughout the entire process (Fig. 1, Supplement Fig. S1), we implemented several elements of control: (1) automatic log files tracking the output of each QC function applied, (2) automatic creation and update of status files, tracking the QC level reached by each dataset, (3) automatic QC summary reports in the form of R markdown documents, (4) interactive Shiny applications for data

visualisation, (5) documentation of manual changes applied to the datasets using manually-edited text files, (6) storage of manual data cleaning operations in text files, and (7) automatic data quality flagging associated with each dataset. All these items ensure a robust, transparent, reproducible and scalable data workflow. Example files for (2), (3) and (6) can be found in the Supplement.

The first stage of the data QC (QC1) performed several data checks (Supplement Table S1) on received spreadsheet files and produced an interactive report in an R markdown document, which signalled possible inconsistencies in the data and warned of potential errors. These data issues were addressed, with the help of data contributors, if needed. Once no errors remained, the dataset was converted into an object of the custom-designed 'sfn_data' class (Supplement Fig. S2, see also section 2.5), which contained all data and metadata for a given dataset (Appendix Tables A2–A6 list all variable names). Data and

metadata belonging to all Level 1 datasets were further visually inspected using an interactive R Shiny application, and, if no major issues were detected, they were subjected to the second QC process, QC2.

## 2.4  Data harmonisation and quality control: QC2

Datasets entering QC2 underwent several data cleaning and data harmonisation processes (Supplement Table S2). We first ran outlier detection and out of range checks; these checks did not delete or modify the data, only warned about any

suspicious observation ('outlier' and 'range' warnings). The outlier detection algorithm was based on a Hampel filter, which also estimates a replacement value for a candidate outlier (Hampel, 1974). For the range checks, we defined minimum and maximum allowed values for all the time series variables, based on published values of extreme weather records and maximum transpiration rates (Cerveny et al., 2007; Manzoni et al., 2013). The outcome of outlier and range checks were visually inspected on the actual time series being evaluated using an interactive R Shiny application (Supplement Fig.S3).

Following expert knowledge, visually confirmed outliers were replaced by the values estimated by the Hampel filter. Similarly, we replaced out of range values by NA if the variable was out of its physically allowed range (Supplement Fig.S3). Outlier and out of range 'warnings' for each observation (e.g. for each variable and timestep) were documented in





two data flags tables, with the same dimensions as the corresponding data tables (Supplement Fig. S2). Likewise, those observations with confirmed problematic values, which were removed or replaced, were also flagged; further information

can be found in the 'data flags' vignettes in the 'sapfluxnetr' package Granda et al. (Granda et al., 2019)

Final data harmonisation processes in QC2 involved unit transformations and the calculation of derived variables (Supplement Table S2). When plant sapwood area was provided by data contributors, we interconverted between sap flow rate per plant and per unit sapwood area. If leaf area was supplied, we also calculated sap flow per unit leaf area, but note

that this transformation does not take into account the seasonal variation in leaf area. In QC2 we estimated missing environmental variables which could be derived from related variables in the dataset (Appendix, Table A6). We also estimated the apparent solar time and extraterrestrial global radiation from the provided timestamp and geographic coordinates using the R package 'solaR' (Perpiñán, 2012). All estimated or interconverted observations were flagged as 'CALCULATED' in the 'env_flags' or 'sap_flags' table (Supplement Fig. S2).

**2.5 Data structure**

One of the major benefits of the SAPFLUXNET data workflow is the encapsulation of datasets in self-contained R objects of the S4 class with a predefined structure. These objects belong to the custom-designed 'sfn_data' class, which display different slots to store time series of sap flow and environmental data, their associated data flags, and all the metadata (Supplement Fig. S2). For further information please see the 'sfn_data classes' vignette in the 'sapfluxnetr' package (Granda

et al., 2019). The code identifying each dataset was created by the combination of a 'country' code, a 'site' code and, if applicable, a 'stand' code and a 'treatment' code. This means that several 'stands' and/or 'treatments' can be present within one 'site' (Supplement Table S3).

At the end of the QC process, we generated a folder structure with a first-level storing datasets as either 'sfn_data' objects or

as a set of comma-separated (csv) text files. Within each of these formats, a second-level folder groups datasets according to how sap flow is normalized (per plant, sapwood or leaf area); note that the same dataset, expressing different sap flow quantities, can be present in more than one folder (e.g. 'plant' and 'sapwood'). Finally, the third level contains the data files for each dataset: either a single 'sfn_data' object storing all data and metadata, or all the individual csv files. More details on the data structure can be found in the 'sapfluxnetr-quick-guide' vignette in the 'sapfluxnetr' package (Granda et al., 2019).





## 3 The SAPFLUXNET database

### 3.1 Data coverage

The SAPFLUXNET version 0.1.5 database harbours 202 globally distributed datasets (Fig. 2a, Supplement Fig. S4 and Table S3), from 121 geographical locations, with Europe, Eastern USA and Australia especially well represented. These datasets were represented in the bioclimatic space using the terrestrial biomes delimited by Whittaker (Fig. 2b), but note that, as any bioclimatic classification, it has its limitations. Datasets have been compiled from all terrestrial biomes, except for temperate rainforests, although some tropical montane sites have been included. Woodland/shrubland and temperate forest biomes are the most represented in the database adding up to 80% of the datasets (Fig. 2b). However, large forested areas in the tropics and in boreal regions are still not well represented (Fig. 2a,b). Looking at the distribution by vegetation type (Fig. 2c), evergreen needleleaf forest is the most represented vegetation type (65 datasets), followed by deciduous broadleaf forest (47 datasets) and evergreen broadleaf forest (43 datasets).

SAPFLUXNET contains sap flow data for 2714 individual plants (1584 angiosperms and 1130 gymnosperms), belonging to 174 species (141 angiosperms and 33 gymnosperms), 95 different genera and 45 different families (Supplement, Table S4-S5). All species but one, *Elaeis guineensis*, a palm, are tree species. *Pinus* and *Quercus* are the most represented genera (Fig. 3b). Amongst the gymnosperms, *Pinus sylvestris*, *Picea abies* and *Pinus taeda* are the three most represented species with data provided on 290, 178 and 107 trees, respectively (Fig. 3a). For the angiosperms, *Acer saccharum*, *Fagus sylvatica* and *Populus tremuloides* are the most represented species, with 162, 116 and 104 trees, respectively, although most *Acer saccharum* data come from a single study with a very large sample size (Fig. 3a). Some species are present in more than 10 datasets: *Pinus sylvestris*, *Picea abies*, *Fagus sylvatica*, *Acer rubrum*, *Liriodendron tulipifera* and *Liquidambar styraciflua* (Fig. 3a, Supplement Table S4).

### 3.2 Methodological aspects

For more than 90% of the plants, sap flow at the whole-plant level is available (either directly provided by contributors or calculated in the QC process); this is important for upscaling SAPFLUXNET data to the stand level (cf. section 4.2). Because the leaf area of the measured plants is often not available as metadata, sap flow per unit leaf area was estimated for only 18.6% of the individuals (Fig. 4). The heat dissipation method is the most frequent method in the database (HD, 66.4% of the plants), followed by the trunk sector heat balance (TSHB, 16.4%) and the compensation heat pulse method (CHP, 8.4%) (Fig. 4). This distribution is broadly similar to the use of each method documented in the literature, although the TSHB method is overrepresented here, compared to the current use of this method by the sap flow community (Flo et al., 2019; Poyatos et al., 2016). Some methods, especially those belonging to the heat pulse family and the cyclic (or transient)



heat dissipation (CHD) method are mostly used in angiosperms, while the TSHB and the heat field deformation (HFD) methods are more frequently used in gymnosperms (Fig. 4).

Calibration of sap flow sensors and scaling from point measurements to the whole-plant can be critical steps towards

accurate estimates of absolute sap flow rates. In SAPFLUXNET, most of the sap flow time series have not undergone a species-specific calibration, with the CHD method showing the highest percentage of calibrated time series (Table 1). This lack of calibrations may be relevant for the more empirical heat dissipation methods (HD and CHD), which have been shown to consistently underestimate sap flow rates (Flo et al., 2019; Peters et al., 2018; Steppe et al., 2010). Radial integration of single-point sap flow measurements is more frequent than azimuthal integration (Table 2), except for the CHD method. A

large number of plants using the HD method, and all plants measured using the HPTM method, do not employ any radial integration procedure. In contrast, the CHP, HR, SHB, and TSHB methods are those which more frequently addressed radial variation in one way or another (Table 2). Azimuthal integration procedures are also more frequent when the TSHB method is used (Table 2).

### 3.3 Plant characteristics

Plant-level metadata is almost complete (99.5% of the individuals) for diameter at breast height (DBH), while sapwood area and sapwood depth, important variables for sap flow upscaling, are not available, or could not be estimated, for 23% and 47% of the plants, respectively. Plant height and plant age are missing for 42% and 62% of the individuals, respectively. Sap flow data in SAPFLUXNET are representative of a broad range of plant sizes (Fig. 5a). The distribution of DBH showed a median of 25.0 cm and 20.4 cm for gymnosperms and angiosperms, respectively, with a long tail towards the largest plants,

two *Mortoniodendron anisophyllum* trees from a tropical forest in Costa Rica that measured > 200 cm (Fig. 5a). The largest gymnosperm tree in SAPFLUXNET (176 cm in DBH) is a kauri tree (*Agathis australis*) from New Zealand. The distribution of plant heights is less skewed, with similar medians for angiosperms (17.6 m) and gymnosperms (17.5 m). The tallest plants are located in a tropical forest in Indonesia, where a *Pouteria firma* tree reached 44.7 m. Remarkably, of the 16 plants taller than 40 m, over 60% are *Eucalyptus* species. The tallest gymnosperm (36.2 m) is a *Pinus strobus* from NE USA.


Plant size metadata in SAPFLUXNET is complemented with plant-level data of sapwood and leaf area, that provide information on the functional areas for water transport and loss (Fig. 5a). Distributions of sapwood and leaf area show highly skewed distributions, with long tails towards the largest values and slightly higher median values for gymnosperms (262 cm$^2$ and 33.0 m$^2$ for sapwood and leaf areas, respectively), compared to angiosperms (168 cm$^2$ and 29.9 m$^2$). Accordingly,

median sapwood depth is also higher for gymnosperms (5.1 cm) compared to angiosperms (3.7 cm). The largest trees (*Mortoniodendron, Pouteria, Agathis*) with deep sapwood (17–24 cm) are also those with largest sapwood areas. Many large angiosperm trees from tropical (CRI_TAM_TOW, IDN_PON_STE, GUF_GUY_ST2; see Table S3 for dataset codes) and





temperate forests (*Fagus grandifolia*, USA_SMIC_SCB) also show large sapwood areas (> 5000 cm$^2$), but the plant with the deepest sapwood is a gymnosperm, an *Abies pinsapo* in Spain with 30.7 cm of sapwood depth.


### 3.4 Stand characteristics

Stand-level metadata include several variables associated with management, vegetation structure and soil properties. Half of the datasets originate from naturally regenerated, unmanaged stands, and 13.9% come from naturally regenerated but managed stands. Plantations add up to 32.2% and orchards only represent 4% of the datasets. Reporting of structural
variables is mixed, with stand height, age, density and basal area showing relatively low missingness (6.4%, 11.4%, 12.9% and 13.4%, respectively); in contrast, soil depth and LAI are missing from 26.7% and 33.7% of the datasets.

SAPFLUXNET datasets originate from stands with diverse structural characteristics. Median stand age is 54 years and there are several datasets coming from >100 year-old forests (Fig. 5b). Stand height shows a similar range and distribution of
values compared to individual plant height (Fig. 5a,b). The denser stands correspond to coppiced evergreen oak stands from Mediterranean forests (FRA_PUE, ESP_TIL_OAK), species-rich tropical forests (MDG_SEM_TAL) or relatively young temperate forests (e.g. FRA_HES_HE1_NON, USA_CHE_MAP). The sparsest stands (< 200 stems ha$^{-1}$) correspond to tree-grass savanna systems (Spain, Portugal, Australia, Senegal), dry woodlands (China), or oil palm plantations in Indonesia (IDN_JAM_OIL). Stands with the largest basal areas (> 70 m$^2$ ha$^{-1}$) are mostly dominated by broadleaf species, except for a
*Picea abies* plantation in Sweden (SWE_SKO_MIN).

The distribution of leaf area index (LAI) shows a median of 3.5 m$^2$ m$^{-2}$, with the largest values observed in temperate (CZE_BIK, USA_DUK_HAR, HUN_SIK) and tropical (GUF_GUY_GUY, COL_MAC_SAF_RAD) forests. The stands with the lowest LAI correspond to the sparse woodlands from Mediterranean and semi-arid locations and also those from
forests near altitudinal or latitudinal tree-lines (FIN_PET, AUT_TSC). SAPFLUXNET datasets show a median soil depth of 100 cm, with only a dozen datasets originated from sites with soils deeper than 10 m (Fig. 5b).

The number of plants per dataset is highly variable, with most of the datasets (86%) containing data for at least 4 trees and 46% of the datasets having data for at least 10 trees (Fig. 6a, see also Fig. 9).

### 3.5 Temporal characteristics

The oldest datasets in SAPFLUXNET go back to 1995 (GBR_DEV_CON, GBR_DEV_DRO) while the most recent data reach up to 2018 (datasets from the ESP_MAJ cluster of sites). Several multi-year datasets are present in SAPFLUXNET (Fig. 6), with 50% of the datasets spanning a period of at least 3 years, and some datasets being extraordinarily long (16 years in FRA_PUE). Frequently, the datasets only cover the 'growing season' periods, or even shorter periods for some sites





which were eventually included because they improved the ecological and geographic coverage of the database (e.g. ARG_MAZ, ARG_TRE as representative of deciduous *Nothofagus* forest in South Patagonia). In contrast, a few datasets show continuous records over multiple years (Fig. 6b). Amongst the longest datasets, most of them come from European or North American sites (Fig. 6), except some datasets from Israel (ISR_YAT_YAT, 7 years), Russia (RUS_FYO, 7 years), South Korea (KOR_TAE cluster of sites, 6 years) or New Zealand (NZL_HUA_HUA, 5 years).


SAPFLUXNET provides an unprecedented database to study the detailed temporal dynamics of plant transpiration across species and sites globally. Sub-daily records of sap flow (e.g. at least at hourly timesteps) are available for extended periods (Fig. 6b), allowing to address both seasonal and diel patterns in water use regulation by trees and how these temporal patterns change across species or years across terrestrial biomes, reflecting different phenologies and water-use strategies.

For instance, in Mediterranean forests, evergreen species such as *Quercus ilex*, *Arbutus unedo* and *Pinus halepensis* show moderate sap flow the whole year round, while the deciduous *Quercus pubescens* shows higher sap flow density during a shorter period and its water use is heavily reduced during a dry year (2012) (Fig. 7a). Temperate forests without water availability limitations show relatively high flows during the growing season and similar diel sap flow patterns among species (Fig. 7b). In contrast, tropical forests show moderate to high sap flow rates during the entire year, with different

dynamics in the intradaily water use regulation across species. For example, *Inga* sp. in a highly diverse wet tropical forest in Costa Rica, reduced sap flow during mid-day hours compared to co-existing species (Fig. 7c).

### 3.6 Availability of environmental data

All SAPFLUXNET datasets contain ancillary time series of the main hydrometeorological drivers of transpiration, accompanied by information on where these variables had been measured (Fig. 8a). Air temperature is available for all

datasets. Although vapour pressure deficit (VPD) was originally absent in 38% of the datasets (Fig. 8a,b), we could estimate it for those sites providing air temperature and relative humidity data (QC Level 2, see section 2.3), and finally only 2 out of the 202 datasets have missing VPD information. For radiation variables, shortwave radiation was most often provided, compared to photosynthetically active and net radiation; only 8 out of 202 datasets do not have any accompanying radiation data. Most of these environmental variables were measured on-site, with precipitation being the variable most frequently

retrieved from nearby meteorological stations (48% of the datasets) (Fig. 8a). Soil water content measured at shallow depth, typically between 0 and 30 cm below the soil surface, is provided for 56% of the datasets, while soil moisture from deep soil layers is available for only 27% of the datasets.



## 4 Potential applications

### 4.1 Applications in plant ecophysiology and functional ecology

There are multiple potential applications of the SAPFLUXNET database to assess whole-plant water use rates and their environmental sensitivity, both across species (e.g. Oren et al., 1999b) and at the intraspecific level (Poyatos et al., 2007). SAPFLUXNET will allow disentangling the roles of evaporative demand and soil water content in controlling transpiration at the plant level, complementing recent studies looking at how water supply and demand affect evapotranspiration at the ecosystem level (Anderegg et al., 2018; Novick et al., 2016). The availability of global sap flow data at sub-daily time
resolution and spanning entire growing seasons will allow focusing on how maximum water use and its environmental sensitivity varies with plant-level attributes such as stem diameter (Dierick and Hölscher, 2009; Meinzer et al., 2005), tree height (Novick et al., 2009; Schäfer et al., 2000), hydraulic (Manzoni et al., 2013; Poyatos et al., 2007) and other plant traits (Grossiord et al., 2019; Kallarackal et al., 2013). SAPFLUXNET thus provides an unprecedented tool to understand how structural and physiological traits scale-up to whole-plant regulation of water fluxes (McCulloh et al., 2019), and how this
integration determines drought responses (Choat et al., 2018) and post-drought recovery patterns (Yin and Bauerle, 2017). Analyses of the temporal dynamics of plant water use in response to specific drought events, as recently assessed for gross primary productivity (e.g. Schwalm et al., 2017), can also help to quantify drought legacy effects, including the reversibility of drought-induced losses of hydraulic conductivity at the plant level.

SAPFLUXNET will allow new insights into within-day patterns and controls in whole-plant water use, which can disclose the fine details of its physiological regulation. Circadian rhythms can modulate stomatal responses to the environment, potentially affecting sap flow dynamics (e.g. de Dios et al., 2015). Hysteresis in diel sap flow relationships with evaporative demand and time-lags between evaporative demand and sap flow, are two linked phenomena likely arising from plant capacitance and other mechanisms (O'Brien et al., 2004; Schulze et al., 1985), that also influence diel evapotranspiration
dynamics (Matheny et al., 2014; Zhang et al., 2014). A major driver of time-lags is the use of stored water to meet the transpiration demand (Phillips et al., 2009), which can now be analysed across species, plant sizes or drought conditions using time series analyses, simplified electric analogies (Phillips et al., 1997, 2004; Ward et al., 2013) or detailed water transport models (Bohrer et al., 2005; Mirfenderesgi et al., 2016). Night-time water use can be substantial for some species (Forster, 2014; Resco de Dios et al., 2019). However, available syntheses rely on study-specific quantification of what
constitutes nocturnal sap flow and do not address possible methodological influences (Zeppel et al., 2014). SAPFLUXNET will allow applying a consistent estimation of nocturnal sap flow and control for datasets that are less suitable for the quantification of night-time fluxes, as information on zero-flow determination is included in the metadata ('pl_sens_cor_zero', Appendix Table A5).





Sap flow data have been widely employed to assess changes in tree water use after biotic (e.g. Hultine et al., 2010) or abiotic (Oren et al., 1999a) disturbances. Likewise, sap flow data have been used to report changes in species and stand water use following experimental treatments involving resource availability modifications (e.g. Ewers et al., 1999) or density changes (i.e. thinning, Simonin et al., 2007). The SAPFLUXNET database includes datasets with experimental manipulations, applied either at the stand or at the individual level (Table 3). The main treatments present are related to thinning, water

availability changes (irrigation, throughfall exclusion) and wildfire impact (Table 3), potentially facilitating new data syntheses and meta-analyses using these datasets (e.g. Grossiord et al., 2017).

The combination of SAPFLUXNET with other ecophysiological databases can inform on the relative sensitivity of different physiological processes in response to drought, for example those related to growth and carbon assimilation (Steppe et al.,

2015) . Within-day fluctuations of stem diameter can be jointly analysed with co-located sap flow measurements to study the dynamics of stored water use under drought and its contribution to transpiration (e.g. Brinkmann et al., 2016), and to infer parameters on tree hydraulic functioning using mechanistic models of tree hydrodynamics (Salomón et al., 2017; Steppe et al., 2006; Zweifel et al., 2007). These analyses could be carried out for a large number of species by combining SAPFLUXNET with data from the Dendroglobal database (http://78.90.202.92/streess/databases/dendroglobal); there are at

least 18 SAPFLUXNET datasets with dendrometer data in Dendroglobal. This database and the International Tree-Ring Data Bank (Zhao et al., 2018) could also be used with SAPFLUXNET to investigate, at the species level, the link between radial growth and water use, including their environmental sensitivity (Morán-López et al., 2014), and how these two processes comparatively respond to drought (Sánchez-Costa et al., 2015). Moreover, given the tight link between water use and carbon assimilation, combining SAPFLUXNET with water-use efficiency from plant $\delta^{13}$C data could potentially be used to estimate

whole-plant carbon assimilation (Hu et al., 2010; Klein et al., 2016; Rascher et al., 2010; Vernay et al., 2020), a quantity that is difficult to measure directly, especially in field-grown, mature trees.

### 4.2 Applications in ecosystem ecology and ecohydrology

SAPFLUXNET will provide a global look at plant water flows to bridge the scales between plant traits and ecosystem fluxes and properties (Reichstein et al., 2014). Vegetation structure, species composition and differential water use strategies

among and within species scale-up to different seasonal patterns of ecosystem transpiration, with a strong influence on ecosystem evapotranspiration and its partitioning. Global controls on evaporative fluxes from vegetation have been mostly addressed using ecosystem (Williams et al., 2012) or catchment evapotranspiration data (Peel et al., 2010). These studies have described global patterns in evapotranspiration driven by different plant functional types or climates, but they cannot be used to quantify and to explain the enormous variation in the regulation of transpiration across and within taxa.


The SAPFLUXNET database will provide a long-demanded data source to be used in ecohydrological research (Asbjornsen et al., 2011). Upscaling individual measurements to the stand level (Čermák et al., 2004; Granier et al., 1996; Köstner et al.,





1998) is necessary to quantitatively compare sap-flow based transpiration with evapotranspiration and transpiration estimates at the ecosystem scale and beyond. Even though SAPFLUXNET was designed to accommodate sap flow data at the plant

level, scaling to the ecosystem level is possible for many datasets. For a basic upscaling exercise using SAPFLUXNET data (Poyatos et al., 2020b), whole-plant sap flow can be normalised by individual basal area (as DBH is usually available in the metadata, cf. section 3.3), averaged for a given species and then scaled to stand level transpiration using total stand basal area and the fraction of basal area occupied by each measured species (see stand metadata, Table A3). For many datasets, sap flow data are available for the species comprising most of the stand basal area (often even 100%, Fig. 9), but species-

based upscaling may be unfeasible in many tropical sites (Fig. 9b), where size-based scaling could be applied instead (e.g. da Costa et al., 2018). Further refinements of the upscaling procedure could be achieved by using trunk diameter distributions of the sap flow plots (Berry et al., 2018). This information, however, is not readily available in SAPFLUXNET, and other data sources (e.g. forest inventories, LIDAR data) or additional simplifying assumptions (i.e. applying the size distribution of measured individuals in the dataset) would be needed.


Stand-level transpiration estimates from a large number of SAPFLUXNET sites can contribute to improve our understanding of the role of forest transpiration in the context of stand water balance and its components at the ecosystem (e.g. Tor-ngern et al., 2018) and catchment levels (Oishi et al., 2010; Wilson et al., 2001). Importantly, SAPFLUXNET can contribute to better understand the global controls on vegetation water use (Good et al., 2017), including the biological and climatic controls on

evapotranspiration partitioning into transpiration and evaporation components (Schlesinger and Jasechko, 2014; Stoy et al., 2019). There is some overlap between the FLUXNET network and SAPFLUXNET (47 datasets from FLUXNET sites). Hence, transpiration from SAPFLUXNET can also be used as a 'ground-truth' reference for transpiration estimates from remote sensing approaches (Talsma et al., 2018) and from eddy covariance data (Nelson et al., accepted). Extrapolating sap flow-derived stand transpiration to large spatial scales can be challenging due to landscape-scale variation in forest structure

(Ford et al., 2007) or topography (Hassler et al., 2018), and to  the low spatial representativeness of sap flow measurements (Mackay et al., 2010). A promising research avenue to help elucidate the role of vegetation in driving hydrological changes across environmental gradients (Vose et al., 2016) would be to combine species-specific stand transpiration data from SAPFLUXNET with stand structural and compositional data from forest inventories (e.g. sapwood area index, Benyon et al., 2015).


Understanding the patterns and mechanisms underlying species interactions with respect to water use within a community is necessary to predict tree species vulnerability to drought (Grossiord, 2019). Multispecies datasets from SAPFLUXNET (Table S4) can be used to assess competition for water resources among species, for example by identifying changes in seasonal water use across co-existing species and hence characterizing the spatiotemporal segregation of their hydrological

niches (Silvertown et al., 2015). By providing a detailed seasonal quantification of tree water use, SAPFLUXNET could also complement isotope-based studies and contribute to interpret the large diversity in root water uptake patterns observed



worldwide (Barbeta and Peñuelas, 2017; Evaristo and McDonnell, 2017) and to explain the different seasonal origin of root-absorbed water across species and environmental gradients (Allen et al., 2019).

Plant water fluxes and hydrodynamics are amongst the most uncertain components of ecosystem and terrestrial biosphere models (Fatichi et al., 2016; Fisher et al., 2018). These models are now incorporating hydraulic traits and processes in their transpiration regulation algorithms (Mencuccini et al., 2019), but multi-site assessments of these algorithms are usually performed against evapotranspiration from eddy flux data (Knauer et al., 2015; Matheny et al., 2014). Model validation against sap flow data has been carried out typically in only one (Kennedy et al., 2019; Williams et al., 2001) or few (Buckley

et al., 2012) sites. SAPFLUXNET can thus contribute to assess the performance of models simulating transpiration of stands or species within stands (e.g. De Cáceres et al., submitted.), for a large number of species and under diverse climatic conditions.

## 5. Limitations and future developments

### 5.1 Limitations

Sap flow data processing differs within and among methods, because different algorithms, calibrations or parameters involved in sap flow calculations may be applied. All of these methods contribute to methodological uncertainty (Looker et al., 2016; Peters et al., 2018) and this challenging methodological variability precludes the implementation of a complete, standardised data workflow from raw to processed data within SAPFLUXNET, as it is done for eddy flux data (Vitale et al., 2020; Wutzler et al., 2018). Commercial software for sap flow data processing from multiple methods is available (i.e. http://

www.sapflowtool.com/SapFlowToolSensors.html) but it has not yet been widely adopted. Freely available data-processing software is only available for the HD method (Oishi et al., 2016; Speckman et al., 2020; Ward et al., 2017).

Sap flow measured with thermometric methods provides a precise estimate of the temporal dynamics of water flow through plants (Flo et al., 2019). However, their performance in measuring absolute flows is mixed. While some well-represented

methods in SAPFLUXNET such as the CHP yield accurate estimates (at least for moderate-to-high flows), the HD method, the most represented method by far, can significantly underestimate water flows (Flo et al., 2019). Because plant-level metadata contain information that document the conversion from raw to processed data (Appendix Table A5), a first-order correction for uncalibrated HD measurements based on available methodological assessments can be applied to allow intercomparability across methods. Nevertheless, given the high unexplained variability (i.e. by species and wood traits) in

the performance of sap flow calibrations (Flo et al., 2019), these corrections should be applied with caution. The determination of zero flow conditions (baselining) can also have significant impacts on the quantification of absolute flow for several methods (Peters et al., 2018; Smith and Allen, 1996; Steppe et al., 2010). The different baselining approaches are also documented in the metadata to inform data syntheses and/or to selectively apply correction factors.




SAPFLUXNET has been designed to store whole-plant sap flow data, and therefore, sap flow measured at multiple points within an individual is not available in the database. Even though this spatial variation could be useful to describe detailed aspects of plant water transport (Nadezhdina et al., 2009), focusing on plant-level data greatly simplifies the data structure. Hence, SAPFLUXNET only includes data already upscaled to the plant level by the data contributors. The main details of how this upscaling process was done for each dataset are provided together with other plant metadata (Table A5), but these

metadata show that within-plant variation in sap flow is often not considered (Table 2). The impact of not accounting for radial and circumferential variability when scaling single-point measurements of sap flow to the whole-plant level can be important (Merlin et al., 2020), but the estimation of sapwood area can also cause large errors (Looker et al., 2016). SAPFLUXNET does not provide information on the method employed to quantify sapwood area (e.g. visual estimation with or without the application of dyes, indirect estimation through allometries at species or site levels) or on the accuracy of

sapwood area data. This precludes uncertainty estimation at the individual level. Future developments in the SAPFLUXNET data structure could include this information as metadata to better document the sensor-to-plant scaling process.

While SAPFLUXNET makes global sap flow data available for the first time, we note that spatial coverage is still sparse and some forested regions are underrepresented in the database (Fig. 2a). We note especially the relatively small number of

datasets for boreal and tropical forests, two important biomes in terms of global water and carbon fluxes (Beer et al., 2010; Schlesinger and Jasechko, 2014). While many geographic gaps are caused by the absence of sap flow studies from such areas, some regions where sap flow studies have been conducted are still not represented in SAPFLUXNET. For example, the recent proliferation of Asian sap flow studies (Peters et al., 2018) has not translated into a high representativity of Asian datasets in SAPFLUXNET yet. Similarly, while the coverage of taxonomic and biometric diversity is unprecedented,

SAPFLUXNET lacks data for the extremely tall trees (Ambrose et al., 2010) or for other growth forms such as shrubs (Liu et al., 2011), lianas (Chen et al., 2015) and other non-woody species (Lu et al., 2002).

### 5.2 Outlook

The public release of SAPFLUXNET has set the stage for a first generation of sap flow-based data syntheses. The work on these syntheses will fuel new ideas and tools for future improvements of the database, as for example new computing

approaches for the processing and analysis of sap flow datasets. One example would be the development of robust imputation algorithms to gap-fill time series of sap flow and environmental data, which can take advantage of tools and datasets already developed by the ecosystem flux community (Moffat et al., 2007; Vuichard and Papale, 2015). The dissemination of SAPFLUXNET will encourage the use of machine-learning algorithms, only occasionally used to analyse sap flow datasets so far (e.g. Whitley et al., 2013). These approaches can also be used to identify the relative importance of

different hydrometeorological drivers of transpiration (Zhao et al., 2019), or to produce global transpiration maps, by combining SAPFLUXNET with other data (Jung et al., 2019). This upscaling of stand transpiration to large areas will also





allow addressing broader questions at the regional and continental scale, such as the role of transpiration in moisture recycling (Staal et al., 2018).

The eventual success of this initiative, in terms of enabling data reuse, contributing towards the understanding and modelling of tree water use at local to global scales will likely encourage the sap flow community to contribute new datasets to future updates of the database. We expect that the development of open-source software for the processing of sap flow raw data (Speckman et al., 2020), its eventual widespread use by the sap flow community and the adoption of standardized calibration practices will increase the quality and intercomparability of future sap flow datasets. These new datasets will hopefully
expand the temporal, geographical and ecological representativity of SAPFLUXNET when new data contribution periods can be opened in the future.

## 6 Data availability, access and feedback

In this paper we present SAPFLUXNET version 0.1.5 (Poyatos et al., 2020a), which contains some small metadata improvements on version 0.1.4, the first one to be made publicly available, in March 2020. Both versions supersede version
0.1.3 which was initially released to data contributors in March 2019. The entire database can be downloaded from its hosting webpage in the Zenodo repository (https://doi.org/10.5281/zenodo.3971689, Poyatos et al. 2020a). In this repository, we provide the database as separate .csv files and as .RData objects; see section 2.4. for details on data structure. Together with the initial publication of SAPFLUXNET in March 2019, we also released the sapfluxnetr R package, available on CRAN, to enable easy access, selection, temporal aggregation and visualisation of SAPFLUXNET data. Feedback on data
quality issues can be forwarded to the SAPFLUXNET initiative email address: sapfluxnet@creaf.uab.cat. All the information about SAPFLUXNET, including the publication of new calls for data contribution, can be found in the project website: http://sapfluxnet.creaf.cat/.

## 7 Conclusions

The SAPFLUXNET database provides the first global perspective of water use by individual plants at multiple timescales,
with important applications in multiple fields, ranging from plant ecophysiology to Earth-system science. This database has been built from community-contributed datasets and is complemented with a software package to facilitate data access. Both the database and the software have been implemented following open science practices, ensuring public access and reproducibility. Data sharing has been a key component of the success of the FLUXNET network of ecosystem fluxes (Bond-Lamberty, 2018), and many databases in plant and ecosystem ecology now offer open data (Bond-Lamberty and
Thomson, 2010; Falster et al., 2015; Gallagher et al., 2020; Kattge et al., 2020). SAPFLUXNET fully aligns with this





philosophy. We expect that this initial data infrastructure will promote data sharing among the sap flow community in the future (Dai et al., 2018) and will allow the continued growth of the SAPFLUXNET database.





**Appendix A: References for individual datasets in SAPFLUXNET**

**Table A1. SAPFLUXNET dataset codes and DOIs (Digital Object Identifiers) of the publications associated with each dataset. When no DOI was available the bibliographic reference is shown. Some datasets may have no associated publication ('unpublished') or they may be listed as 'under review'.**

| site_code | DOI |
| --- | --- |
| ARG_MAZ | https://doi.org/10.1007/s00468-013-0935-4 |
| ARG_TRE | https://doi.org/10.1007/s00468-013-0935-4 |
| AUS_BRI_BRI | unpublished |
| AUS_CAN_ST1_EUC | https://doi.org/10.1016/j.foreco.2009.07.036 |
| AUS_CAN_ST2_MIX | https://doi.org/10.1016/j.foreco.2009.07.036 |
| AUS_CAN_ST3_ACA | https://doi.org/10.1016/j.foreco.2009.07.036 |
| AUS_CAR_THI_00F | https://doi.org/10.1016/j.foreco.2011.11.019 |
| AUS_CAR_THI_0P0 | https://doi.org/10.1016/j.foreco.2011.11.019 |
| AUS_CAR_THI_0PF | https://doi.org/10.1016/j.foreco.2011.11.019 |
| AUS_CAR_THI_CON | https://doi.org/10.1016/j.foreco.2011.11.019 |
| AUS_CAR_THI_T00 | https://doi.org/10.1016/j.foreco.2011.11.019 |
| AUS_CAR_THI_T0F | https://doi.org/10.1016/j.foreco.2011.11.019 |
| AUS_CAR_THI_TP0 | https://doi.org/10.1016/j.foreco.2011.11.019 |
| AUS_CAR_THI_TPF | https://doi.org/10.1016/j.foreco.2011.11.019 |
| AUS_ELL_HB_HIG | https://doi.org/10.1016/j.jhydrol.2015.02.045 |
| AUS_ELL_MB_MOD | https://doi.org/10.1016/j.jhydrol.2015.02.045 |
| AUS_ELL_UNB | https://doi.org/10.1016/j.jhydrol.2015.02.045 |
| AUS_KAR | unpublished |
| AUS_MAR_HSD_HIG | https://doi.org/10.1002/eco.1463 |
| AUS_MAR_HSW_HIG | https://doi.org/10.1002/eco.1463 |
| AUS_MAR_MSD_MOD | https://doi.org/10.1002/eco.1463 |
| AUS_MAR_MSW_MOD | https://doi.org/10.1002/eco.1463 |
| AUS_MAR_UBD | https://doi.org/10.1002/eco.1463 |
| AUS_MAR_UBW | https://doi.org/10.1002/eco.1463 |
| AUS_RIC_EUC_ELE | https://doi.org/10.1111/1365-2435.12532 |
| AUS_WOM | https://doi.org/10.1016/j.foreco.2016.12.017 |
| AUT_PAT_FOR | https://doi.org/10.1007/s10342-013-0760-8 |
| AUT_PAT_KRU | https://doi.org/10.1007/s10342-013-0760-8 |
| AUT_PAT_TRE | https://doi.org/10.1007/s10342-013-0760-8 |
| AUT_TSC | https://doi.org/10.10167j.flora.2014.06.012 |
| BRA_CAM | https://doi.org/10.1093/treephys/tpv001 |
| BRA_CAX_CON | https://doi.org/10.1111/gcb.13851 |
| BRA_SAN | https://doi.org/10.1016/j.agrformet.2012.02.002; https://doi.org/10.1007/s00468-015-1165-8; https://doi.org/10.1007/s00468-017-1527-5 |
| BRA_SAN | https://doi.org/10.1016/j.agrformet.2012.02.002; https://doi.org/10.1007/s00468-015-1165-8; https://doi.org/10.1007/s00468-017-1527-5 |
| CAN_TUR_P39_POS | https://doi.org/10.1016/j.agrformet.2010.04.008; https://doi.org/10.1002/hyp.9315 |
| CAN_TUR_P39_PRE | https://doi.org/10.1016/j.agrformet.2010.04.008; https://doi.org/10.1002/hyp.9315 |
| CAN_TUR_P74 | https://doi.org/10.1016/j.agrformet.2010.04.008 |
| CHE_DAV_SEE | https://doi.org/10.1007/s10021-011-9481-3 |
| CHE_LOT_NOR | https://doi.org/10.1111/pce.13500 |



| site_code | DOI |
|---|---|
| CHE_PFY_CON | https://doi.org/10.1093/treephys/tpp123 |
| CHE_PFY_IRR | https://doi.org/10.1093/treephys/tpp123 |
| CHN_ARG_GWD | https://doi.org/10.1016/j.foreco.2016.08.049 |
| CHN_ARG_GWS | https://doi.org/10.1016/j.foreco.2016.08.049 |
| CHN_HOR_AFF | https://doi.org/10.5194/bg-2017-69 |
| CHN_YIN_ST1 | https://doi.org/10.1016/j.foreco.2016.08.049 |
| CHN_YIN_ST2_DRO | https://doi.org/10.1016/j.foreco.2016.08.049 |
| CHN_YIN_ST3_DRO | https://doi.org/10.1016/j.foreco.2016.08.049 |
| CHN_YUN_YUN | https://doi.org/10.5194/bg-11-5323-2014 |
| COL_MAC_SAF_RAD | unpublished |
| CRI_TAM_TOW | https://doi.org/10.1002/hyp.10960 |
| CZE_BIK | unpublished |
| CZE_BIL_BIL | unpublished |
| CZE_KRT_KRT | unpublished |
| CZE_LAN | unpublished;https://doi.org/10.1098/rstb.2019.0518 |
| CZE_LIZ_LES | https://doi.org/10.2136/vzj2012.0154 |
| CZE_RAJ_RAJ | https://doi.org/10.3832/ifor1307-007 |
| CZE_SOB_SOB | https://doi. org/10.14214/sf.1760 |
| CZE_STI | unpublished |
| CZE_UTE_BEE | unpublished |
| CZE_UTE_BNA | unpublished |
| CZE_UTE_BPO | unpublished |
| CZE_UTE_SPR | unpublished |
| DEU_HIN_OAK | unpublished;https://doi.org/10.2136/vzj2018.06.0116 |
| DEU_HIN_TER | unpublished;https://doi.org/10.2136/vzj2018.06.0116 |
| DEU_MER_BEE_NON | https://doi.org/10.4432/0300-4112-86-83 |
| DEU_MER_BEE_THI | https://doi.org/10.4432/0300-4112-86-83 |
| DEU_MER_DOU_NON | https://doi.org/10.4432/0300-4112-86-83 |
| DEU_MER_DOU_THI | https://doi.org/10.4432/0300-4112-86-83 |
| DEU_MER_MIX_NON | https://doi.org/10.4432/0300-4112-86-83 |
| DEU_MER_MIX_THI | https://doi.org/10.4432/0300-4112-86-83 |
| DEU_STE_2P3 | https://doi.org/10.1051/forest:2007020; https://doi.org/10.3390/f11050537 |
| DEU_STE_4P5 | https://doi.org/10.1051/forest:2007020; https://doi.org/10.3390/f11050537 |
| ESP_ALT_ARM | https://doi.org/10.1007/s11258-014-0351-x; https://doi.org/10.1093/treephys/tpy022; https://doi.org/10.1016/j.envexpbot.2018.08.006; https://doi.org/10.1016/j.agwat.2012.06.024 |
| ESP_ALT_HUE | https://doi.org/10.1007/s11258-014-0351-x |
| ESP_ALT_TRI | unpublished; https://doi.org/10.1007/s10342-013-0687-0 |
| ESP_CAN | https://doi.org/10.1016/j.agrformet.2015.03.012 |
| ESP_GUA_VAL | https://doi.org/10.1093/jxb/erw121; https://doi.org/10.1093/treephys/tpw029 |
| ESP_LAH_COM | https://doi.org/10.1007/s00271-015-0471-7 |
| ESP_LAS | https://doi.org/10.1007/s10342-014-0779-5; https://doi.org/10.1016/j.agrformet.2014.11.008 |
| ESP_MAJ_MAI | Perez-Priego et al., under review |
| ESP_MAJ_NOR_LM1 | https://doi.org/10.1016/j.agrformet.2017.01.009 |



| site_code | DOI |
|---|---|
| ESP_MON_SIE_NAT | https://doi.org/10.1016/j.agwat.2012.06.024; https://doi.org/10.1016/0378-1127(96)03729-2; https://doi.org/10.1007/s004680050229; https://doi.org/10.1016/j.actao.2004.01.003; https://doi.org/10.1007/s11258-004-7007-1; https://doi.org/10.1093/treephys/25.8.1041; https://doi.org/10.1007/s00468-007-0192-5; https://doi.org/10.1111/pce.12103 |
| ESP_RIN | https://doi.org/10.1016/j.foreco.2008.03.004 |
| ESP_RON_PIL | https://doi.org/10.3390/f10121132 |
| ESP_SAN_A_45I | https://doi.org/10.1007/s11104-013-1704-2; https://doi.org/10.1016/j.agwat.2012.06.027; https://doi.org/10.1016/j.agrformet.2015.11.013 |
| ESP_SAN_A2_45I | https://doi.org/10.1007/s11104-013-1704-2; https://doi.org/10.1016/j.agwat.2012.06.027; https://doi.org/10.1016/j.agrformet.2015.11.013 |
| ESP_SAN_B_100 | https://doi.org/10.1007/s11104-013-1704-2; https://doi.org/10.1016/j.agwat.2012.06.027; https://doi.org/10.1016/j.agrformet.2015.11.013 |
| ESP_SAN_B2_100 | https://doi.org/10.1007/s11104-013-1704-2; https://doi.org/10.1016/j.agwat.2012.06.027; https://doi.org/10.1016/j.agrformet.2015.11.013 |
| ESP_TIL_MIX | https://doi.org/10.1111/nph.12278 |
| ESP_TIL_OAK | https://doi.org/10.3390/f6082505;https://doi.org/10.1111/nph.12278 |
| ESP_TIL_PIN | https://doi.org/10.3390/f6082505;https://doi.org/10.1111/nph.12278 |
| ESP_VAL_BAR | https://doi.org/10.1093/treephys/27.4.537;https://doi.org/10.5194/hess-9-493-2005 |
| ESP_VAL_SOR | https://doi.org/10.5194/hess-9-493-2005;https://doi.org/10.1016/j.agrformet.2007.05.003 |
| ESP_YUN_C1 | https://doi.org/10.1016/j.foreco.2017.10.017;https://doi.org/10.3390/f10121132 |
| ESP_YUN_C2 | https://doi.org/10.1016/j.foreco.2017.10.017;https://doi.org/10.3390/f10121132 |
| ESP_YUN_T1_THI | https://doi.org/10.1016/j.foreco.2017.10.017;https://doi.org/10.3390/f10121132 |
| ESP_YUN_T3_THI | https://doi.org/10.1016/j.foreco.2017.10.017;https://doi.org/10.3390/f10121132 |
| FIN_HYY_SME | https://doi.org/10.1007/978-94-007-5603-8_9 |
| FIN_PET | unpublished;https://doi.org/10.1016/j.agrformet.2012.02.009 |
| FRA_FON | https://doi.org/10.1111/nph.13771 |
| FRA_HES_HE1_NON | https://doi.org/10.1051/forest:2008052 |
| FRA_HES_HE2_NON | https://doi.org/10.1051/forest:2008052 |
| FRA_PUE | https://doi.org/10.1111/j.1365-2486.2009.01852.x |
| GBR_ABE_PLO | https://doi.org/10.1111/j.1365-3040.2007.01647.x. |
| GBR_DEV_CON | https://doi.org/10.1093/treephys/18.6.393 |
| GBR_DEV_DRO | https://doi.org/10.1093/treephys/18.6.393 |
| GBR_GUI_ST1 | https://doi.org/10.1007/s00442-006-0552-7 |
| GBR_GUI_ST2 | https://doi.org/10.1007/s00442-006-0552-7 |
| GBR_GUI_ST3 | https://doi.org/10.1007/s00442-006-0552-7 |
| GUF_GUY_GUY | https://doi.org/10.1111/j.1365-2486.2008.01610.x |
| GUF_GUY_ST2 | https://doi.org/10.1111/j.1744-7429.2012.00902.x |
| GUF_NOU_PET | https://doi.org/10.1111/1365-2435.13188 |
| HUN_SIK | Mészáros, I., Kanalas, P., Fenyvesi, A., Kis, J., Nyitrai, B., Szollosi, E., Oláh, V., Demeter, Z., Lakatos, Á., & Ander, I. (2011). Diurnal and seasonal changes in stem radius increment and sap flow density indicate different responses of two co-existing oak species to drought stress. Acta Silvatica et Lignaria Hungarica, 7, 97-108. |
| IDN_JAM_OIL | https://doi.org/10.1016/j.agrformet.2019.04.017; http://doi:10.1093/treephys/tpv013 |
| IDN_JAM_RUB | https://doi.org/10.1016/j.agrformet.2019.04.017; https://doi.org/10.1002/eco.1882 |
| IDN_PON_STE | https://doi.org/10.1007/s13595-011-0110-2 |
| ISR_YAT_YAT | https://doi.org/10.1111/nph.13597 |
| ITA_FEI_S17 | https://doi.org/10.1111/nph.15348 |
| ITA_KAE_S20 | https://doi.org/10.1111/nph.15348 |





| site_code | DOI |
| --- | --- |
| ITA_MAT_S21 | https://doi.org/10.1111/nph.15348 |
| ITA_MUN | https://doi.org/10.1111/nph.15348 |
| ITA_REN | unpublished |
| ITA_RUN_N20 | https://doi.org/10.1111/nph.15348 |
| ITA_TOR | https://doi.org/10.1007/s00484-012-0614-y;https://doi.org/10.1007/s00484-008-0152-9 |
| JPN_EBE_HYB | unpublished |
| JPN_EBE_SUG | unpublished |
| KOR_TAE_TC1_LOW | https://doi.org/10.1007/s10310-014-0463-0 |
| KOR_TAE_TC2_MED | https://doi.org/10.1007/s10310-014-0463-0 |
| KOR_TAE_TC3_EXT | https://doi.org/10.1007/s10310-014-0463-0 |
| MDG_SEM_TAL | unpublished |
| MDG_YOU_SHO | https://doi.org/10.1093/treephys/tpy004 |
| MEX_COR_YP | https://doi.org/10.1016/j.agrformet.2013.11.002;https://doi.org/10.1016/j.agrformet.2012.08.004 |
| MEX_VER_BSJ | unpublished |
| MEX_VER_BSM | unpublished |
| NLD_LOO | https://doi.org/10.1016/j.agrformet.2011.07.020 |
| NLD_SPE_DOU | https://doi.org/10.17026/dans-zvq-dq4w |
| NZL_HUA_HUA | unpublished;https://doi.org/10.1007/s00468-015-1164-9 |
| PRT_LEZ_ARN | https://doi.org/10.1002/hyp.10097 |
| PRT_MIT | https://doi.org/10.1093/treephys/27.6.793 |
| PRT_PIN | https://doi.org/10.1007/s10021-011-9453-7 |
| RUS_CHE_LOW | https://doi.org/10.1002/eco.2132 |
| RUS_CHE_Y4 | https://doi.org/10.1002/2016JG003709 |
| RUS_FYO | unpublished;https://doi.org/10.3402/tellusb.v54i5.16679 |
| RUS_POG_VAR | https://doi.org/10.1016/j.agrformet.2019.02.038; https://doi.org/10.17660/ActaHortic.2018.1222.17 |
| SEN_SOU_IRR | https://doi.org/10.1093/treephys/28.1.95 |
| SEN_SOU_POS | https://doi.org/10.1093/treephys/28.1.95 |
| SEN_SOU_PRE | https://doi.org/10.1093/treephys/28.1.95 |
| SWE_NOR_ST1_AF1 | https://doi.org/10.1016/S0168-1923(99)00092-1 |
| SWE_NOR_ST1_AF2 | https://doi.org/10.1016/S0168-1923(99)00092-1 |
| SWE_NOR_ST1_BEF | https://doi.org/10.1016/S0168-1923(99)00092-1 |
| SWE_NOR_ST2 | https://doi.org/10.1016/S0168-1923(99)00092-1 |
| SWE_NOR_ST3 | https://doi.org/10.1016/S0168-1923(99)00092-1 |
| SWE_NOR_ST4_AFT | https://doi.org/10.1016/j.foreco.2007.12.047 |
| SWE_NOR_ST4_BEF | https://doi.org/10.1016/j.foreco.2007.12.047 |
| SWE_NOR_ST5_REF | https://doi.org/10.1016/j.foreco.2007.12.047 |
| SWE_SKO_MIN | https://doi.org/10.1139/cjfr-2016-0541 |
| SWE_SKY_38Y | unpublished |
| SWE_SKY_68Y | unpublished |
| SWE_SVA_MIX_NON | https://doi.org/10.5194/hess-24-2999-2020 |
| THA_KHU | https://doi.org/10.1093/treephys/tpr058 |
| USA_BNZ_BLA | https://doi.org/10.1002/2014JG002683 |
| USA_CHE_ASP | https://doi.org/10.1029/2007WR006272; https://doi.org/10.1029/2009WR008125; https://doi.org/:10.1029/2009JG001092; https://doi.org/10.1111/j.1365-2435.2009.01657.x |
| USA_CHE_MAP | https://doi.org/10.1111/j.1365-2435.2009.01657.x;https://doi.org/10.1029/2009WR008125;https://doi.org/10.1029/2010JG001377 |





| site_code | DOI |
| --- | --- |
| USA_DUK_HAR | https://doi.org/10.1016/j.agrformet.2008.06.013 |
| USA_HIL_HF1_POS | https://doi.org/10.1002/hyp.10474 |
| USA_HIL_HF1_PRE | https://doi.org/10.1002/hyp.10474 |
| USA_HIL_HF2 | https://doi.org/10.1002/hyp.10474 |
| USA_HUY_LIN_NON | https://doi.org/10.2307/3858565 |
| USA_INM | https://doi.org/10.1016/S0168-1923(00)00199-4;https://doi.org/10.1046/j.1365-2486.2002.00492.x |
| USA_MOR_SF | https://doi.org/10.1093/treephys/tpw126 |
| USA_NWH | https://doi.org/10.1002/2015JG003208 |
| USA_ORN_ST1_AMB | https://doi.org/10.1093/treephys/tpr002;https://doi.org/10.1002/eco.173 |
| USA_ORN_ST2_AMB | https://doi.org/10.1093/treephys/tpr002;https://doi.org/10.1002/eco.173 |
| USA_ORN_ST3_ELE | https://doi.org/10.1002/eco.173 |
| USA_ORN_ST4_ELE | https://doi.org/10.1002/eco.173 |
| USA_PAR_FER | https://doi.org/10.1111/j.1469-8137.2010.03245.x;https://doi.org/10.1111/j.1365-3040.2009.01981.x;http://doi.org/10.5849/forsci.11-051 |
| USA_PER_PER | https://doi.org/10.3390/f7100214 |
| USA_PJS_P04_AMB | http://doi.org/10.1890/ES11-00369.1 |
| USA_PJS_P08_AMB | http://doi.org/10.1890/ES11-00369.1 |
| USA_PJS_P12_AMB | http://doi.org/10.1890/ES11-00369.1 |
| USA_SIL_OAK_1PR | https://doi.org/10.1002/hyp.10104;https://doi.org/ 10.1111/j.1365-2486.2009.02037.x;https://doi.org/10.1093/treephys/tpt122 |
| USA_SIL_OAK_2PR | https://doi.org/10.1002/hyp.10104;https://doi.org/ 10.1111/j.1365-2486.2009.02037.x;https://doi.org/10.1093/treephys/tpt122 |
| USA_SIL_OAK_POS | https://doi.org/10.1002/hyp.10104;https://doi.org/ 10.1111/j.1365-2486.2009.02037.x;https://doi.org/10.1093/treephys/tpt122 |
| USA_SMI_SCB | https://doi.org10.1111/1365-2435.12470 |
| USA_SMI_SER | unpublished;https://doi.org/10.1002/ece3.1117 |
| USA_SWH | https://doi.org/10.1002/2015JG003208 |
| USA_SYL_HL1 | https://doi.org/10.1029/2005JG000083 |
| USA_SYL_HL2 | https://curate.nd.edu/show/hm50tq60r1c |
| USA_TNB | https://doi.org/10.1016/S0168-1923(00)00199-4 |
| USA_TNO | https://doi.org/10.1016/S0168-1923(00)00199-4 |
| USA_TNP | https://doi.org/10.1016/S0168-1923(00)00199-4 |
| USA_TNY | https://doi.org/10.1016/S0168-1923(00)00199-4 |
| USA_UMB_CON | https://doi.org/10.1002/2014JG002804 |
| USA_UMB_GIR | https://doi.org/10.1002/2014JG002804 |
| USA_WIL_WC1 | https://doi.org/10.1016/j.agrformet.2004.06.008 |
| USA_WIL_WC2 | unpublished |
| USA_WVF | https://doi.org/10.1016/S0168-1923(00)00199-4;https://doi.org/10.1016/S0168-1923(96)02375-1 |
| UZB_YAN_DIS | https://doi.org/10.1016/j.foreco.2007.09.005 |
| ZAF_FRA_FRA | https://doi.org/10.1016/j.foreco.2015.11.009 |
| ZAF_NOO_E3_IRR | https://doi.org/10.1016/j.agrformet.2019.02.042;Gush, M.B., Dzikiti, S., Clulow, A.D., Mengistu, M.G., Jarmain, C., Taylor, N.J. and Everson, C.S. 2014. Water use of apple orchards. In: Gush, M.B. and Taylor, N.J. (Eds) 2014. The water use of selected fruit tree orchards (Volume 2): Technical report on measurements and modelling. Water Research Commission Report No.1770/2/14, Section 3. WRC, Pretoria, RSA. (ISBN 978-1-4312-0575-2). |





| site_code | DOI |
|---|---|
| ZAF_RAD | https://doi.org/10.1016/j.agwat.2018.06.017;https://doi.org/10.17159/wsa/2020.v46.i2.8236 |
| ZAF_SOU_SOU | https://doi.org/10.1016/j.agwat.2018.06.017;https://doi.org/10.17159/wsa/2020.v46.i2.8236 |
| ZAF_WEL_SOR | https://doi.org/10.1016/j.foreco.2017.05.009 |




**Table A2.** Description of site metadata variables in SAPFLUXNET datasets.

| Variable | Description | Type | Units |
|---|---|---|---|
| si_name | Site name given by contributors | Character | None |
| si_country | Country code (ISO) | Character | Fixed values |
| si_contact_firstname | Contributor first name | Character | None |
| si_contact_lastname | Contributor last name | Character | None |
| si_contact_email | Contributor email | Character | None |
| si_contact_institution | Contributor affiliation | Character | None |
| si_addcontr_firstname | Additional contributor first name | Character | None |
| si_addcontr_lastname | Additional contributor last name | Character | None |
| si_addcontr_email | Additional contributor email | Character | None |
| si_addcontr_institution | Additional contributor affiliation | Character | None |
| si_lat | Site latitude (i.e. 42.36) | Numeric | Latitude, decimal format (WGS84) |
| si_long | Site longitude (i.e. -8.23) | Numeric | Longitude, decimal format (WGS84) |
| si_elev | Elevation above sea level | Numeric | meters |
| si_paper | Paper with relevant information on the dataset, as DOI links or DOI codes | Character | DOI link |
| si_dist_mgmt | Recent and historic disturbance and management events that affected the measurement years | Character | Fixed values |
| si_igbp | Vegetation type based on IGBP classification | Character | Fixed values |
| si_flux_network | Logical indicating if site is participating in the FLUXNET network | Logical | Fixed values |
| si_dendro_network | Logical indicating if site is participating in the DENDROGLOBAL network | Logical | Fixed values |
| si_remarks | Remarks and commentaries useful to grasp some site-specific peculiarities | Character | None |
| si_code | Sapfluxnet site code, unique for each site | Character | Fixed value |
| si_mat | Site annual mean temperature, as obtained from CHELSA | Numeric | Celsius degrees |
| si_map | Site annual mean precipitation, as obtained from CHELSA | Numeric | mm |
| si_biome | Biome classification based on Whittaker (1970) , based on MAT and MAP obtained from CHELSA. | Character | sapfluxnet calculated |





**Table A3.** Description of stand metadata variables in SAPFLUXNET datasets.

| Variable | Description | Type | Units |
|---|---|---|---|
| st_name | Stand name given by contributors | Character | None |
| st_growth_condition | Growth condition with respect to stand origin and management | Character | Fixed values |
| st_treatment | Treatment applied at stand level | Character | None |
| st_age | Mean stand age at the moment of sap flow measurements | Numeric | years |
| st_height | Canopy height | Numeric | meters |
| st_density | Total stem density for stand | Numeric | stems/ha |
| st_basal_area | Total stand basal area | Numeric | $m^2$/ha |
| st_lai | Total maximum stand leaf area (one-sided, projected) | Numeric | $m^2/m^2$ |
| st_aspect | Aspect the stand is facing (exposure) | Character | Fixed values |
| st_terrain | Slope and/or relief of the stand | Character | Fixed values |
| st_soil_depth | Soil total depth | Numeric | cm |
| st_soil_texture | Soil texture class, based on simplified USDA classification | Character | Fixed values |
| st_sand_perc | Soil sand content, % mass | Numeric | % percentage |
| st_silt_perc | Soil silt content, % mass | Numeric | % percentage |
| st_clay_perc | Soil clay content, % mass | Numeric | % percentage |
| st_remarks | Remarks and commentaries useful to grasp some stand-specific peculiarities | Character | None |
| st_USDA_soil_texture | USDA soil classification based on the percentages provided by the contributor | Character | sapfluxnet calculated |



**Table A4.** Description of species metadata variables in SAPFLUXNET datasets.

| Variable | Description | Type | Units |
|---|---|---|---|
| sp_name | Identity of each measured species | Character | Scientific name without author abbreviation, as accepted by The Plant List |
| sp_ntrees | Number of trees measured of each species | Numeric | number of trees |
| sp_leaf_habit | Leaf habit of the measured species | Character | Fixed values |
| sp_basal_area_perc | Basal area occupied by each measured species, in percentage over total stand basal area | Numeric | % percentage |



**Table A5.** Description of plant metadata variables in SAPFLUXNET datasets.

| Variable | Description | Type | Units |
|---|---|---|---|
| pl_name | Plant code assigned by contributors | Character | None |
| pl_species | Species identity of the measured plant | Character | Scientific name without author abbreviation, as accepted by The Plant List |
| pl_treatment | Experimental treatment (if any) | Character | None |
| pl_dbh | Diameter at breast height of measured plants | Numeric | cm |
| pl_height | Height of measured plants | Numeric | m |
| pl_age | Plant age at the moment of measure | Numeric | years |
| pl_social | Plant social status | Character | Fixed values |
| pl_sapw_area | Cross-sectional sapwood area | Numeric | $cm^2$ |
| pl_sapw_depth | Sapwood depth, measured at breast height | Numeric | cm |
| pl_bark_thick | Plant bark thickness | Numeric | mm |
| pl_leaf_area | Leaf area of each measured plant | Numeric | $m^2$ |
| pl_sens_meth | Sap flow measures method | Character | Fixed values |
| pl_sens_man | Sap flow measures sensor manufacturer | Character | Fixed values |
| pl_sens_cor_grad | Correction for natural temperature gradients method | Character | Fixed values |
| pl_sens_cor_zero | Zero flow determination method | Character | Fixed values |
| pl_sens_calib | Was species-specific calibration used? | Logical | Fixed values |
| pl_sap_units | Sapfluxnet-harmonised units for sap flow at the sapwood, leaf and plant level | Character | Fixed values |
| pl_sap_units_orig | Original sap flow units provided by the contributors | Character | Fixed values |
| pl_sens_length | Length of the needles or electrodes forming the sensor | Numeric | mm |
| pl_sens_hgt | Sensor installation height, measured from the ground | Numeric | m |
| pl_sens_timestep | Subdaily time step of sensor measures | Numeric | minutes |
| pl_radial_int | Integration of radial variation in sap flow along sapwood depth | Character | Fixed values |
| pl_azimut_int | Integration of azimuthal variation of sap flow along stem circumference | Character | Fixed values |
| pl_remarks | Remarks and commentaries useful to grasp some plant-specific peculiarities | Character | None |
| pl_code | Sapfluxnet plant code, unique for each plant | Character | Fixed value |





**Table A6.** Description of environmental metadata variables in SAPFLUXNET datasets.

| Variable | Description | Type | Units |
|---|---|---|---|
| env_time_zone | Time zone of site used in the TIMESTAMPS | Character | Fixed values |
| env_time_daylight | Is daylight saving time applied to the original timestamp? | Logical | Fixed values |
| env_timestep | Sub-daily timestep of environmental measurements | Numeric | minutes |
| env_ta | Location of air temperature sensor | Character | Fixed values |
| env_rh | Location of relative humidity sensor | Character | Fixed values |
| env_vpd | Location of vapour pressure deficit measurements | Character | Fixed values |
| env_sw_in | Location of shortwave incoming radiation sensor | Character | Fixed values |
| env_ppfd_in | Location of incoming photosynthetic photon flux density sensor | Character | Fixed values |
| env_netrad | Location of net radiation sensor | Character | Fixed values |
| env_ws | Location of wind speed sensor | Character | Fixed values |
| env_precip | Location of precipitation measurements | Character | Fixed values |
| env_swc_shallow_depth | Average depth for shallow soil water content measures | Numeric | cm |
| env_swc_deep_depth | Average depth for deep soil water content measures | Numeric | cm |
| env_plant_watpot | Availability of water potential values for the same measured plants during the sap flow measurements period | Character | Fixed values |
| env_leafarea_seasonal | Availability of seasonal course of leaf area data | Character | Fixed values |
| env_remarks | Remarks and commentaries useful to grasp some environmental-specific peculiarities | Character | None |

**Author contributions**

V. Granda, R. Poyatos, V. Flo and J. Martínez-Vilalta designed and built the database. RP, VG, and VF summarised the database and drafted the manuscript, with the contribution of JMV, M. Mencuccini and K.Steppe. The rest of coauthors contributed data to the database and edited the manuscript.



**Competing interests**

The authors declare that they have no conflict of interest.

**Acknowledgements**

This data compilation would have not been possible without the contribution of all the people who supported the construction and maintenance of measurement infrastructures, helped with field data collection and participated in data processing of individual datasets. We would also like to acknowledge the support of Agustí Escobar, Roberto Molowny-

Horas, Marie Sirot and Guillem Bagaria in building the data infrastructure.

**Financial support**

The data compilation and the building of the data infrastructure involved in this research has been supported by the grants CGL2014-55883-JIN, RTI2018-095297-J-I00 (MINECO and MICINN, Spain) and CAS16/00207 (MECD, Spain), by the grant SGR1001 (AGAUR, Catalonia) and by a Humboldt Research Fellowship for Experienced Researchers (Germany). VF

has been supported by the doctoral fellowship FPU15/03939 (MECD, Spain). JMV benefited from an ICREA Academia award.

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





**Tables**


**Table 1. Number of sap flow times series in SAPFLUXNET depending on whether they were calibrated (species-specific), non-calibrated or this information was not provided, for the different sap flow methods: cyclic (or transient) heat dissipation (CHD), compensation heat pulse (CHP), heat dissipation (HD), heat field deformation (HFD), heat pulse T-max (HPTM), heat ratio (HR), stem heat balance (SHB) and trunk sector heat balance (TSHB). The**
**percentage of calibrated time series was expressed with respect to the total number of sap flow time series for each method.**

| Method | Calibrated | Non-calibrated | Not provided | % calibrated |
|--------|-----------|----------------|--------------|--------------|
| CHD | 6 | 13 | 0 | 31.6 |
| CHP | 29 | 42 | 157 | 12.7 |
| HD | 214 | 1491 | 98 | 11.9 |
| HR | 3 | 55 | 47 | 2.9 |
| TSHB | 7 | 433 | 4 | 1.6 |
| HFD | 0 | 8 | 0 | 0.0 |
| HPTM | 0 | 80 | 0 | 0.0 |
| SHB | 0 | 27 | 0 | 0.0 |



**Table 2. Number of plants in the SAPFLUXNET database using different radial and azimuthal integration approaches for the different sap flow methods: cyclic (or transient) heat dissipation (CHD), compensation heat pulse (CHP), heat dissipation (HD), heat field deformation (HFD), heat pulse T-max (HPTM), heat ratio (HR), stem heat balance (SHB) and trunk sector heat balance (TSHB).**

| Azimuthal integration | | | | | |
|---|---|---|---|---|---|
| Method | Measured | Sensor-integrated | Corrected, measured azimuthal variation | No azimuthal correction | Not provided |
| CHD | 15 | 0 | 0 | 0 | 4 |
| CHP | 61 | 0 | 0 | 167 | 0 |
| HD | 216 | 0 | 520 | 1021 | 46 |
| HFD | 0 | 0 | 0 | 8 | 0 |
| HPTM | 0 | 0 | 0 | 80 | 0 |
| HR | 7 | 0 | 2 | 88 | 8 |
| SHB | 0 | 0 | 0 | 27 | 0 |
| TSHB | 0 | 25 | 191 | 219 | 9 |
| Radial integration | | | | | |
| Method | Measured | Sensor-integrated | Corrected, measured radial variation | No radial correction | Not provided |
| CHD | 0 | 0 | 6 | 13 | 0 |
| CHP | 222 | 0 | 6 | 0 | 0 |
| HD | 77 | 3 | 645 | 703 | 142 |
| HFD | 2 | 0 | 0 | 6 | 0 |
| HPTM | 0 | 0 | 0 | 80 | 0 |
| HR | 57 | 1 | 42 | 3 | 2 |
| SHB | 0 | 27 | 0 | 0 | 0 |
| TSHB | 0 | 338 | 8 | 89 | 9 |



**Table 3. Number of datasets, plants and species by stand-level treatment in the SAPFLUXNET database.**

| Treatment | N sites | N plants | N species |
|---|---|---|---|
| None/control | 155 | 2198 | 170 |
| Thinning | 18 | 332 | 18 |
| Irrigation | 9 | 36 | 4 |
| Post-fire | 6 | 18 | 4 |
| $CO_2$ fertilisation | 3 | 28 | 2 |
| Drought | 3 | 9 | 2 |
| Soil fertilisation | 2 | 16 | 2 |
| Post-mortality | 1 | 22 | 5 |
| Soil fertilisation and pruning | 1 | 12 | 1 |
| Soil fertilisation and thinning | 1 | 12 | 1 |
| Pruning and thinning | 1 | 11 | 1 |
| Soil fertilisation, pruning and thinning | 1 | 11 | 1 |
| Pruning | 1 | 9 | 1 |




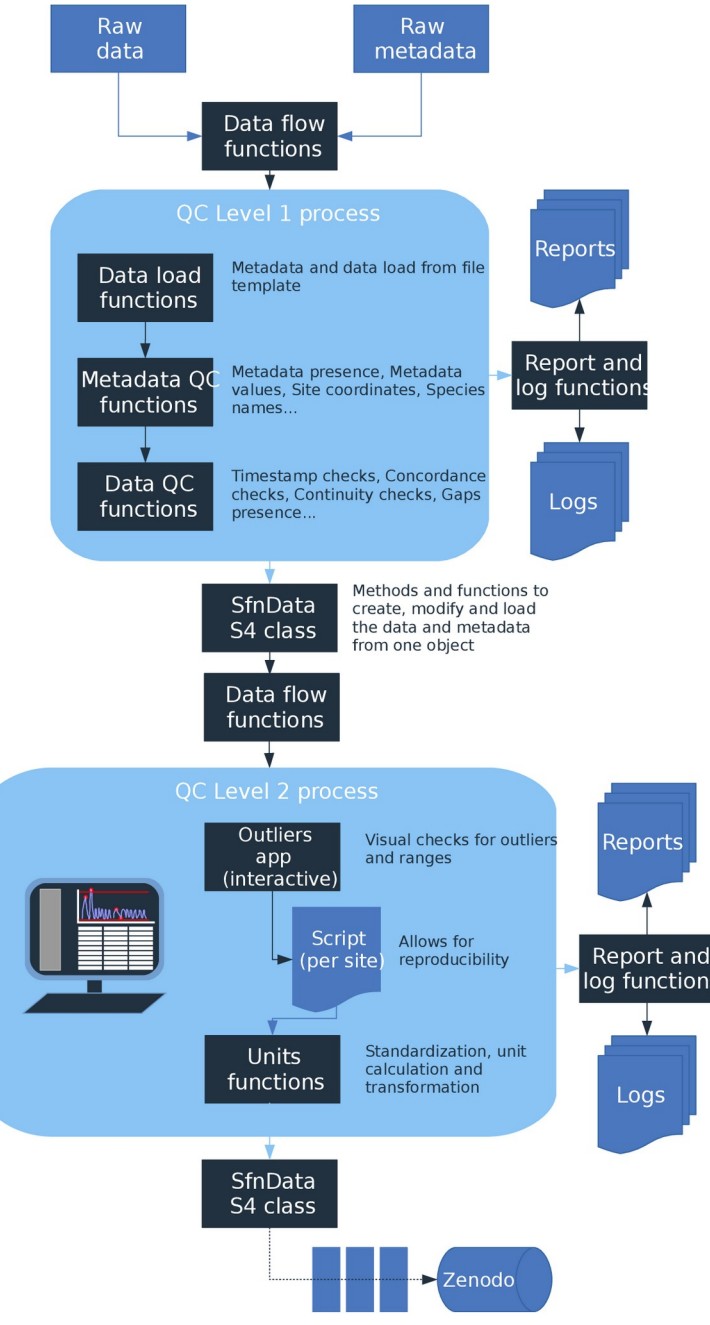

**Figure 1. Overview of the SAPFLUXNET data workflow. Data files are received from data contributors, and undergo several quality-control processes (QC1 and QC2). Both, QC1 and QC2 produce an .RData object of the custom-designed sfn-data S4 class storing all data, metadata and data flags for each dataset. The progress and results of the QC processes are monitored through individual reports and log files. The final outcome, is stored in a folder structure with a either single .RData file for each dataset or a set of seven csv files for each dataset.**




**Figure 2. (a) Geographic, (b) bioclimatic and (c) vegetation type distribution of SAPFLUXNET datasets. In (a) woodland area**
**from Crowther et al. (2015) is shown in green. In (b) we represent the different datasets according to their mean annual**
**temperature and precipitation in a Whittaker diagram showing the classification of the main terrestrial biomes. In (c) vegetation**
**types are defined according to the International Geosphere-Biosphere Programme (IGBP) classification (ENF: Evergreen**
**Needleaf Forest; DBF: Deciduous Broadleaf Forest; EBF: Evergreen Broadleaf Forest; MF: Mixed Forest; DNF: Deciduous**
**Needleleaf forest; SAV: Savannas; WSA: Woody Savannas; WET: Permanent Wetlands).**







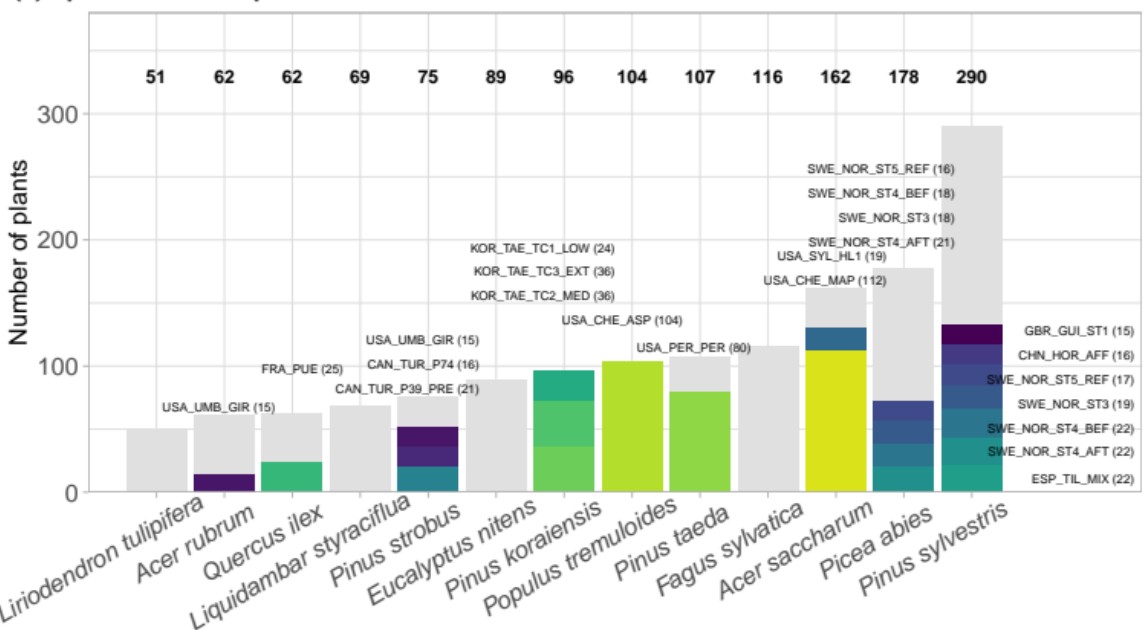

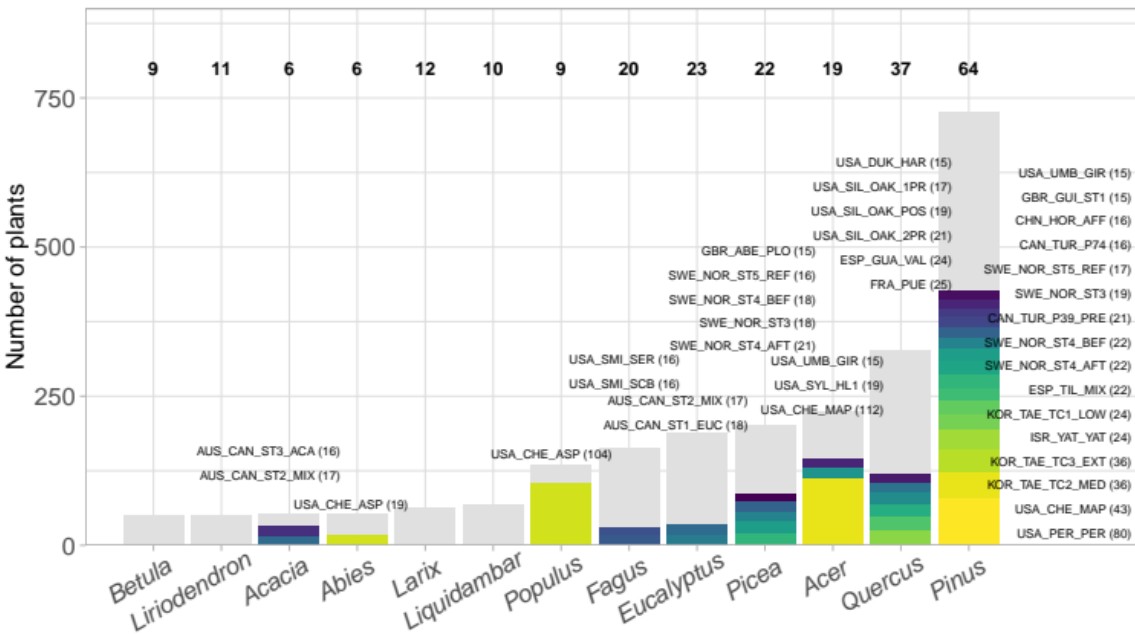

**Figure 3. Taxonomic distribution of genera and species in SAPFLUXNET, showing (a) species and (b) genera with > 50 plants in the database. Total bar height depicts number of plants per species (a) or genera (b). Numbers on top of each bar show the number of datasets where each species (a) or genus (b) is present. Colours other than grey highlight datasets with 15 or more plants of a given species (a) or genus (b). Bar height for a given colour is proportional to the number of plants in the corresponding dataset, which is also shown in parentheses next to the dataset code.**




**Figure 4. Distribution of plants in SAPFLUXNET according to major taxonomic group (angiosperms, gymnosperms), sap flow method (CHD:cycling heat dissipation; CHP: compensation heat pulse; HD: heat dissipation; HFD: heat field deformation: HPTM: heat pulse T-max (HPTM): HRM: heat ratio (HR); SHB: stem heat balance; TSHB: trunk sector heat balance) and reference unit for the expression of sap flow (plant, sapwood area, leaf area). Combinations of reference units imply that data are present in multiple units.**




## (a) Plant attributes

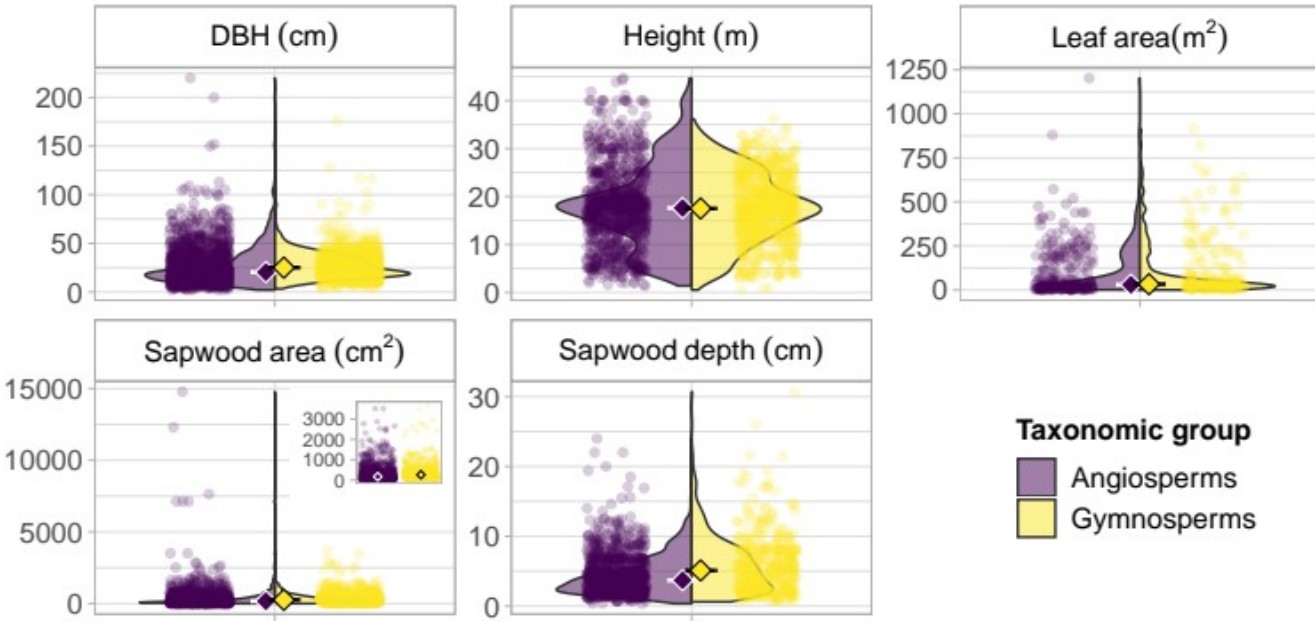

## (b) Stand attributes

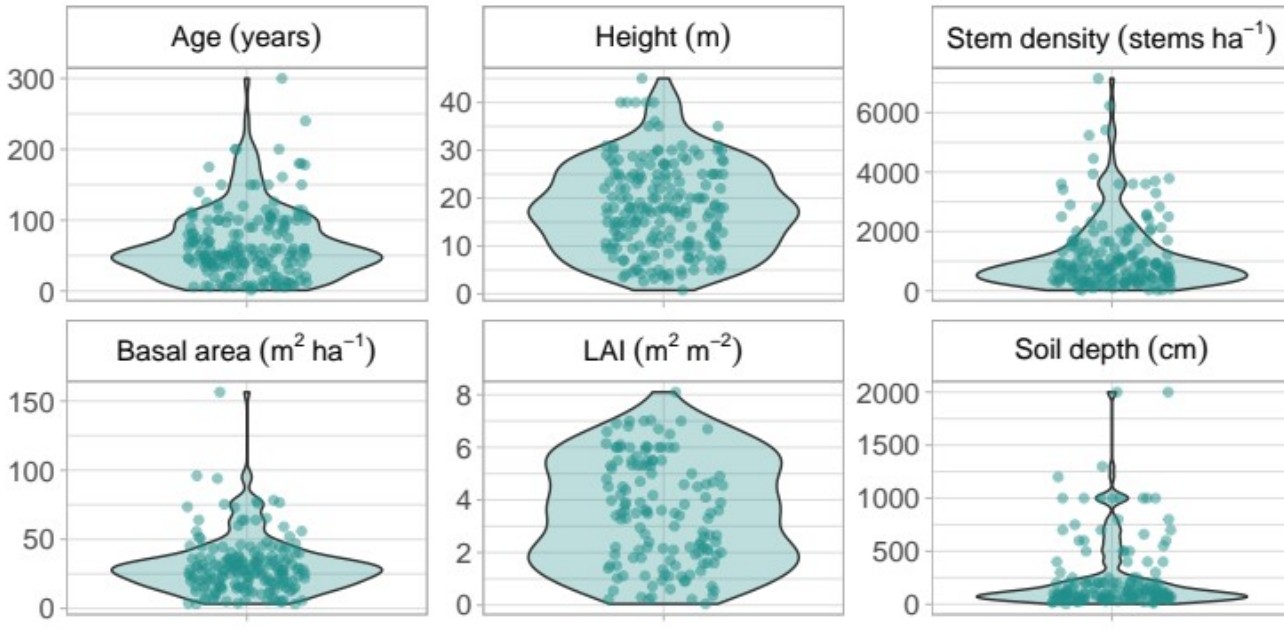

**Figure 5. Characteristics of trees and stands in the SAPFLUXNET database.** Panel (a) shows plant data and kernel density plots of the main plant attributes, coloured by taxonomic group (angiosperms and gymnosperms): diameter at breast height (DBH), plant height, sapwood area, sapwood depth and leaf area. The inset in the sapwood area panel zooms in values lower than 5000 cm². Panel (b) shows stand data and kernel density plots of the main stand attributes: stand age, stand height, stem density, stand basal area,leaf area index (LAI) and soil depth.



**Figure 6. (a)** Measurement duration of SAPFLUXNET datasets expressed in number of days with sap flow data and coloured by the number of plants measured on each day . The 30 longest datasets are labelled. For each dataset in panel (a), panel (b) shows its corresponding measurement period.





**Figure 7. Fingerprint plots showing hourly sap flow per unit sapwood area (colour scale) as a function of hour of day (x-axis) and day of year (y-axis) for a selection of SAPFLUXNET sites with at least four co-occurring species. Panel (a) shows data from a Woodland/Shrubland forest in NE Spain (ESP_CAN), for an average (2011) and a dry (2012) year. Panel (b) shows data for a mesic Temperate forest (USA_WVF) and panel (c) shows data for a Tropical forest (CRI_TAM_TOW). For this latter site, only 4 of the 17 measured species are shown and some of them were only identified at the genus level.**






**Figure 8. Summary of the availability of different environmental variables in SAPFLUXNET datasets. (a) Distribution of**
**meteorological variables according to sensor location (in brackets, names of the variables in the database), (b) Distribution of soil**
**moisture variables according to the measurement depth (in brackets, names of the variables in the database). (c) Venn diagram**
**showing the number of datasets where each combination of different environmental variables are present, grouping shortwave,**
**PPFD and net radiation under 'Radiation' variables.**




**Figure 9. Potential for upscaling species-specific plant sap flow to stand-level sap flow using SAPFLUXNET datasets. Datasets are**
**shown using an aggregated biome classification; 'Dry and Tropical' include: 'Subtropical desert', 'Temperate grassland**



desert', 'Tropical forest savanna' and 'Tropical rain forest'. Each panel shows the percentage of total stand basal area that is covered by sap flow measurements for each species in the dataset. Datasets are also coloured by the number of species present. Numbers on top of each bar depict the total number of plants for a given dataset. Empty bars show datasets for which sap flow data expressed at the plant level were not available.
