# Peer review of "Global transpiration data from sap flow measurements: the SAPFLUXNET database"

_Earth System Science Data, 2020_

## Referee Comment (RC1) · Stan Schymanski (Referee) · 16 Nov 2020

GENERAL COMMENTS

The manuscript entitled "Global transpiration data from sap flow measurements: the SAPFLUXNET database" is a great addition to ESSD, and I would like to thank the authors for making such a large and helpful dataset public. It is also really good to see that a lot of the software used for data processing, quality control and subsequent analysis is also publicly available, but this is not described very well in the Data availability section (Section 6). In fact, I was not able to entirely understand the data harmonisation and quality control described in Section 2.4 (see some of my detailed comments below), and without the accompanying scripts, it may be difficult to reproduce the data

in the database from the original data, should questions emerge.

The database itself is very large and it is a pity that the whole database has to be downloaded and de-compressed before inspecting it, but this is a limitation of the zenodo repository. To make access to the data more convenient, the authors could also place a copy on github or gitlab, where users would be able to download individual files or folders. I was also confused about the organisation of the data into 'leaf', 'plant', and 'sapwood'. If a dataset contained information about leaf area and sapwood area, would it then be repeated in all three folders? This would not seem very efficient, as it would unnecessarily inflate the database in my eyes.

I went through the Quick Guide (http://sapfluxnet.creaf.cat/sapfluxnetr/articles/sapfluxnetr-quick-guide.html) and found that working with the database in R is relatively easy even for readers with very limited experience with R. What I found a little confusing is that the names of the columns in 'arg_maz_sapf' are not explained anywhere, and the units are also not readily accessible, although they appear correctly in the plots. It would be helpful if the Quick Guide explained how to find out what the column names mean and what the units are. It would also be helpful to include code that would allow easy reproduction of the figures in the paper and, ideally, also figures in the original papers linked to each dataset. This would help the reader to better understand the intended use of the data. Perhaps the latter could be followed for the next version of the database.

The manuscript contains a very inspiring section on potential uses of the database. However, in this section and other places in the manuscript, it seems to equate sapflow dynamics with whole plant transpiration dynamics, which can be quite misleading, given capacity-caused delays between those two variables, which are mentioned in passing but not explicitly discussed in the context of shifts in timing between transpiration and sapflow. See detailed comments below.

I share the editor's concern that data uncertainty is not adequately addressed in the

[Figure]

paper and in the data base. The manuscript discusses various aspects of uncertainty, cites evidence for consistent underestimation of sapflow rates by some methods and suggests that a first-order correction is possible, but it does not provide any numbers or details about the correction. Similarly, it mentions that estimation of sapwood area can have substantial errors without providing numbers and even an attempt to estimate the propagation of such errors to whole plant sap flow rates. I find that this is a missed opportunity, given the exquisite selection of co-authors on this manuscript. It would be great if the authors could give some guidance to the readers on the likely error bounds of the data in the database.

Below, I will answer to the ESSD review criteria one-by-one.

Are the data and methods presented new? I believe that the database and underlying methods are cutting edge, as described in the manuscript.

Is there any potential of the data being useful in the future? Definitely, as described in the manuscript, with the caveat that it should be clarified that the diurnal dynamics of sapflow should not be expected to be the same as that of transpiration, and subject to some quantification of potential error / uncertainty.

Are methods and materials described in sufficient detail? Mostly yes, but I was not able to easily understand some of them, as pointed out above.

Are any references/citations to other data sets or articles missing or inappropriate? Not that I am aware of.

Is the article itself appropriate to support the publication of a data set? Yes, it is very detailed and honest.

Is the data set accessible via the given identifier? Yes.

Is the data set complete? I did not really check, but a colleague of mine has been using the data more intensively and has not found any issues.
Are error estimates and sources of errors given (and discussed in the article)? Unfortunately, no quantitative error estimates were given, although sources of errors are discussed in the article. Please see above.

Are the accuracy, calibration, processing, etc. state of the art? I could not find any issues, but I am also much less qualified to assess this than the many co-authors on this manuscript.

Are common standards used for comparison? Unfortunately, I am not aware of any common standards for comparison against sapflow data. Lysimeter studies might be helpful, but I did not see any such studies cited in the manuscript.

Is the data set significant – unique, useful, and complete? I am convinced it is, except for error estimates.

Are there any inconsistencies, implausible assertions or noticeable problems that would suggest the data are erroneous (or worse)? Most datasets included in the database have already been used and published elsewhere but I did not assess whether the results of the original publications can be reproduced using the database, nor did I see such an analysis as part of the data quality checks. Given that most of the analysis is done with R, it would have been great if the authors included code to reproduce the figures and tables presented in this manuscript, and, perhaps for future data submissions, to reproduce figures in the original publications related to each dataset.

Is the data set itself of high quality? The quality checks on the data were quite systematic and the metadata included is also very helpful, instigating trust in the quality of the data.

Is the data set usable in its current format and size? Yes.

Are the formal metadata appropriate? Yes.

Is the length of the article appropriate? Yes.
Is the overall structure of the article well structured and clear? Yes.

Is the language consistent and precise? Yes, with a few exceptions, see detailed comments below.

Are mathematical formulae, symbols, abbreviations, and units correctly defined and used? I did not find any problems.

Are figures and tables correct and of high quality? Since the code used to generate the figures and tables in the manuscript was not provided, I was not able to verify within the time I had for this review.

Is the data set publication, as submitted, of high quality? Yes.

Finally: By reading the article and downloading the data set, would you be able to understand and (re-)use the data set in the future? Very likely.

DETAILED COMMENTS

L290: "understanding of"

L295: This sentence does not make much sense to me: "In practice, transpiration is relatively easy to isolate from the bulk evaporative flux, evapotranspiration, only from the leaf to the plant levels." First of all, it is not easy to separate transpiration from ET at the leaf level, as the only other component of ET at leaf level is evaporation of intercepted water, which is not easily separated from transpiration. Secondly, transpiration measurement at the plant level is exactly what sapflow is about, so why are you saying that it is relatively easy?

L314: When stating that sap flow sensors can be deployed in almost any terrestrial ecosystem, it would be important to point out that they are limited to woody plants (unless I am wrong).

L338: Where are the quality assurance and control procedures described?
L364: What do you mean by "methodological uncertainty"? Uncertainty about what method was used for a given dataset due to inadequate documentation, or uncertainty about what is the most suitable method for a given case? Could you clarify in the text?

L372: This sounds as if the stem heat balance method was superior to the others. Could you clarify?

L432-434: What does the Hampel filter do? How does it identify outliers and estimate correct values? Where is the code used to determine out of range values and outliers? Where is this R Shiny application? I could not find it at https://github.com/sapfluxnet. What is the role of expert knowledge here? Why were out of range values replaced by NA and outliers by the Hampel filter value?

L440: I shink the first "Granda et al." should be removed.

L445: Was sap flow per unit leaf area computed even for deciduous species using constant leaf area all year round? Does it make sense to do this?

L450-465: These details could be moved to the appendix, as I did not find it very informative without seeing the database.

L490: Grammar: A sentence should not start with "Because".

L503: By how much did HD and CHD underestimate sap flow rates?

L505: Grammar: reads as if plants were doing the data processing.

L510: What does it mean for the upscaling? Could it not be done at all for datasets with missing sapwood area or depth?

L529: Is there any reliable way to estimate sapwood area for trees where this information was not provided? I did not understand what role tree height plays for the use and/or interpretation of the data.

L583: What does the "compared to..." mean? PAR and Rnet were provided less often

than shortwave? Was shortwave converted to PAR or vice versa to obtain a homogenous dataset?

L594: It would be good to mention that sub-daily sap flow variation is not necessarily consistent with sub-daily variation in transpiration.

L599: What do you mean by "scale up to"? Maybe "translate to"?

L504: How could the data be used to estimate hydraulic conductivity?

L605-610: This may be misleading, as diurnal patterns in sap flow do not necessarily translate to diurnal patterns in transpiration, due to capacitance. So the relevant time lags are not only between evaporative demand and sap flow, as stated here, but also between sap flow and transpiration. This needs to be clarified to avoid confusion.

L616: Why would SAPFLUXNET allow quantification of nocturnal sap flow for data sets less suitable for quantification of night-time fluxes? More explanation would be helpful here.

L625: How are treatments documented in the database? Qualitatively or quantitatively?

L658: Has any of this been attempted already, and if so, how big are the uncertanties?

L676: To know if this up-scaling is promising, it would be important to know the uncertainty in sap flow derived individual-based or sapwood area based transpiration.

L704: How could uncertainty propagation ever be done for the commercial methods? This might be a good place to emphasise the need for open source data-processing software.

L710-711: How accurate, how big is the under-estimation?

L711: Grammar: don't start a sentence with "Because"

L715: So what error bands should we expect for the data before and after corrections?

L728: How large can the errors due to sapwood area estimation errors be?

L773: Could you add the relevant packages used for data processing and quality control, some of which are mentioned in https://github.com/sapfluxnet, and specivy which version of each package was used?

---

## Referee Comment (RC2) · Robert Skelton (Referee) · 7 Jan 2021

General comments: The SAPFLUXNET database represents an excellent undertaking by Poyatos et al., and these authors should be congratulated for a fine achievement. The paper is well laid out and clear and I only have a few minor comments for the authors that I hope will improve the manuscript. It will be good to see this paper/dataset published and I look forward to being able to contribute to it and to make use of it in the future. One suggestion for an improvement is to highlight an example of where the dataset has been used to answer an experimental question. For example, is there a dataset that can demonstrate the impact of an abiotic stress on sap flow for different species? Specific comments: Line 290: The main line of inquiry is not how plants regulate transpiration, but rather how transpiration varies with abiotic factors and along

environmental gradients in different species. Perhaps the wording could be changed to something like "An improved understanding of transpiration and how variable this process is under different abiotic conditions, along environmental gradients and in different species is thus needed to. . ." Line 312: Several studies have also quantified sap flow of graminoid species; see for example Skelton et al. 2012 (There are other papers). Line 313: See Clearwater et al. 2009 for a non-invasive approach using non-invasive external probes. Also cite Clearwater et al. 2009 on line 351. References 1. Skelton, R. P., West, A. G., Dawson, T. E. & Leonard, J. M. External heat-pulse method allows comparative sapflow measurements in diverse functional types in a Mediterranean-type shrubland in South Africa. Funct. Plant Biol. 40, 1076–1087 (2013). 2. Clearwater, M. J., Luo, Z., Mazzeo, M. & Dichio, B. An external heat pulse method for measurement of sap flow through fruit pedicels, leaf petioles and other small-diameter stems. Plant. Cell Environ. 32, 1652–1663 (2009).

Line 361: It would be best to avoid acronyms to make the sentence (and other sentences) easier to read. It is much easier to read "heat dissipation or compensation heat pulse" in this sentence than to keep referring to where the acronym was first mentioned. Line 405: Are there plans for this site to be maintained? Can people continue to contribute? Perhaps state this up front.

---

## Author Comment (AC1) · 3 Mar 2021

Please see the supplement file for a formatted version of this document (comments and replies in different font and style). The document contains a table and full captions for the figures, as the space provided in the submission application is limited.

R1#1. GENERAL COMMENTS The manuscript entitled "Global transpiration data from sap flow measurements: the SAPFLUXNET database" is a great addition to ESSD, and I would like to thank the authors for making such a large and helpful dataset public. It is also really good to see that a lot of the software used for data processing, quality control and subsequent analysis is also publicly available, but this is not described very well in the Data availability section (Section 6). In fact, I was not able to entirely understand

the data harmonisation and quality control described in Section 2.4 (see some of my detailed comments below), and without the accompanying scripts, it may be difficult to reproduce the data in the database from the original data, should questions emerge.

Thanks for your positive comments. All the data processing of the originally-submitted data spreadsheets (henceforth 'data spreadsheets') is available in the sapfluxnetQC1 R package. This process is described in the vignettes associated to this package (https://github.com/sapfluxnet/sapfluxnetQC1/tree/master/vignettes); we will add a reference to these vignettes in section 2.3:

All R code involved in this QC process was implemented in the sapfluxnetQC1 package (Granda et al., 2016); see the package vignettes for a detailed description (https://github.com/sapfluxnet/sapfluxnetQC1/tree/master/vignettes).

On the reviewer's concerns on reproducibility, it should be possible for us to rebuild the entire database from the spreadsheets in our own server. Our goal with this strategy was to facilitate the internal data processing for us, and in fact we have sometimes had to rebuild the database from the data spreadsheets when substantial changes had to be made in the data processing pipeline. However, this reproducibility has been designed for internal data processing in our sever, not for its general application.

R1#2. The database itself is very large and it is a pity that the whole database has to be downloaded and de-compressed before inspecting it, but this is a limitation of the zenodo repository. To make access to the data more convenient, the authors could also place a copy on github or gitlab, where users would be able to download individual files or folders.

A direct programmatic access from R or other languages would be desirable, but as you say, this is a Zenodo limitation. However, Zenodo has several other benefits, including the handling of different versions of the data (DOI versioning).

As for the access to the data, once downloaded, we provide a very handy set of tools in

the sapfluxnetr package to access, filter, aggregate and visualize data. With regard to the gihub/gitlab integration, these are really software platforms, and to the best of our knowledge they are not designed as data repositories and don't really support storing large amounts of data.

R1#3. I was also confused about the organisation of the data into 'leaf', 'plant', and 'sapwood'. If a dataset contained information about leaf area and sapwood area, would it then be repeated in all three folders? This would not seem very efficient, as it would unnecessarily inflate the database in my eyes.

Yes, a dataset with sap flow expressed per leaf, plant and sapwood area would be repeated in the three folders. We are aware that it's not the most efficient solution in terms of data storage, and we opted for it in part because data storage is less of a problem nowadays. Moreover, it's a simple way to organise the data and allowing the users straightforward access to data according to their needs (e.g. plant-level data for ecosystem-level upscaling).

R1#5. I went through the Quick Guide (http://sapfluxnet.creaf.cat/sapfluxnetr/articles/sapfluxnetrquick-guide.html) and found that working with the database in R is relatively easy even for readers with very limited experience with R. What I found a little confusing is that the names of the columns in 'arg_maz_sapf' are not explained anywhere, and the units are also not readily accessible, although they appear correctly in the plots. It would be helpful if the Quick Guide explained how to find out what the column names mean and what the units are.

The quick guide (http://sapfluxnet.creaf.cat/sapfluxnetr/articles/sapfluxnetr-quick-guide.html) already shows how to use function in sapfluxnetr called 'describe_md_variable' which precisely details values and units for metadata variables. There is also a dedicated section in the web page where information on data and metadata variables can be found:

http://sapfluxnet.creaf.cat/sapfluxnetr/articles/metadata-and-data-units.html

To clarify this issue, we have added further information in the text on variables and units, at the end of section 2.5:

More details on the data structure and units can be found in the 'sapfluxnetr-quick-guide' and 'metadata-and-data-units' vignettes, respectively, in the 'sapfluxnetr' package (Granda et al., 2019).

And at the end of section 2.3:

Once no errors remained, the dataset was converted into an object of the custom-designed 'sfn_data' class (Supplement Fig. S2, see also section 2.5), which contained all data and metadata for a given dataset (Appendix Tables A2–A6 list all variable names and units).

R1#6. It would also be helpful to include code that would allow easy reproduction of the figures in the paper and, ideally, also figures in the original papers linked to each dataset. This would help the reader to better understand the intended use of the data. Perhaps the latter could be followed for the next version of the database

The code to reproduce the figures in this paper is available in the following Github repository: https://github.com/sapfluxnet/sfn_datapaper. This is now stated in the code availability section.

As for the comment on the figures in the original papers associated to the datasets, we think that it would require a lot of effort in trying to exactly match the particularities of the figures of each specific study, and, to us, the benefits of this exercise are not really clear.

R1#7. The manuscript contains a very inspiring section on potential uses of the database. However, in this section and other places in the manuscript, it seems to equate sapflow dynamics with whole plant transpiration dynamics, which can be quite misleading, given capacity-caused delays between those two variables, which are mentioned in passing but not explicitly discussed in the context of shifts in timing between

transcription and sapflow. See detailed comments below

The reviewer is right, we have modified one sentence in the introduction to reflect this:

Xylem sap flow can be upscaled to the whole plant, obtaining a near-continuous quantification of plant water use, keeping in mind that stem sap flow typically lags behind canopy transpiration (Schulze et al. 1985).

This adds to the explicit consideration of sap flow-transpiration time-lags in section 4.1:

Hysteresis in diel sap flow relationships with evaporative demand and time-lags between evaporative demand and sap flow, are two linked phenomena likely arising from plant capacitance and other mechanisms (O'Brien et al., 2004; Schulze et al., 1985), that also influence diel evapotranspiration dynamics (Matheny et al., 2014; Zhang et al., 2014). A major driver of time-lags is the use of stored water to meet the transpiration demand (Phillips et al., 2009), which can now be analysed across species, plant sizes or drought conditions using time series analyses, simplified electric analogies (Phillips et al., 1997, 2004; Ward et al., 2013) or detailed water transport models (Bohrer et al., 2005; Mirfenderesgi et al., 2016).

R1#8. I share the editor's concern that data uncertainty is not adequately addressed in the paper and in the data base. The manuscript discusses various aspects of uncertainty, cites evidence for consistent underestimation of sapflow rates by some methods and suggests that a first-order correction is possible, but it does not provide any numbers or details about the correction. Similarly, it mentions that estimation of sapwood area can have substantial errors without providing numbers and even an attempt to estimate the propagation of such errors to whole plant sap flow rates. I find that this is a missed opportunity, given the exquisite selection of co-authors on this manuscript. It would be great if the authors could give some guidance to the readers on the likely error bounds of the data in the database.

We agree that estimation of uncertainties could have been more thoroughly addressed

in the manuscript. We also want to stress that the availability of global sap flow data in SAPFLUXNET will precisely stimulate the development of new frameworks to quantify and upscale uncertainties, as occurred with the compilation of global eddy flux data in the FLUXNET network. Here is the new text in section 5.1:

Overall, this first global compilation of sap flow data will allow addressing uncertainties in sap flow upscaling in space and time in the same way that the development of FLUXNET stimulated the quantification and aggregation of uncertainties for eddy flux data (Richardson et al. 2012).

Richardson, A. D., Aubinet, M., Barr, A. G., Hollinger, D. Y., Ibrom, A., Lasslop, G. and Reichstein, M.: Uncertainty Quantification, in Eddy Covariance: A Practical Guide to Measurement and Data Analysis, edited by M. Aubinet, T. Vesala, and D. Papale, pp. 173–209, Springer Netherlands, Dordrecht, https://doi.org/10.1007/978-94-007-2351-1_7, , 2012.

In the revised version we have added a new section 3.7 dealing with uncertainty estimation and specific methodological corrections.

Based on the data in the global meta-analysis of sap flow calibration by Flo et al. 2019, we provide examples and code (see the README at https://github.com/sapfluxnet/sfn_datapaper) for uncertainty estimations associated with each of the main sap flow density methods, together with an example application of heat dissipation bias-correction (Appendix B, Table B1, Fig. B1-B3, see the appendix at the end of the document). We have also published the raw data from the calibrations in a Zenodo repository (Flo et al. 2021; https://zenodo.org/record/4559497#.YD5ibXVKiis), cited in the appendix. This way, other uncertainty quantifications can be based in these data, which constitute the only global compilation of sap flow data comparisons against reference measurements.

We also compare methodological uncertainty with those derived from sapwood area estimation, when upscaling to whole-plant sap flow rates (Appendix B, Fig. B3b,c see

the appendix at the end of this document).

Finally, we apply wood type- specific radial profiles of sap flow density to sap flow measurements that were not radially-integrated by data contributor, to show the potential impact of scaling assumptions that don't consider radial integration of sap flow (Appendix B, Fig. B4 see the appendix at the end of this document). Here's the text for section 3.7:

[revised manuscript text omitted]

R1#9. Below, I will answer to the ESSD review criteria one-by-one. Are the data and methods presented new? I believe that the database and underlying methods are cutting edge, as described in the manuscript Are any references/citations to other data sets or articles missing or inappropriate? Not that I am aware of. Is the article itself appropriate to support the publication of a data set? Yes, it is very detailed and honest. Is the data set accessible via the given identifier? Yes. Is the data set complete? I did not really check, but a colleague of mine has been using the data more intensively and has not found any issues. Are the accuracy, calibration, processing, etc. state of the art? I could not find any issues, but I am also much less qualified to assess this than the many co-authors on this manuscript. Is the data set significant – unique, useful, and complete? I am convinced it is, except for error estimates. Is the data set itself of high quality? The quality checks on the data were quite systematic and the metadata included is also very helpful, instigating trust in the quality of the data. Is the data set usable in its current format and size? Yes. Are the formal metadata appropriate? Yes. Is the length of the article appropriate? Yes. Is the overall structure of the article well structured and clear? Yes. Is the language consistent and precise? Yes, with a few exceptions, see detailed comments below. Are mathematical formulae, symbols, abbreviations, and units correctly defined and used? I did not find any problems.

Thanks for this careful assessment of our paper and for the positive comments.

R1#10. Is there any potential of the data being useful in the future? Definitely, as described in the manuscript, with the caveat that it should be clarified that the diurnal

dynamics of sapflow should not be expected to be the same as that of transpiration, and subject to some quantification of potential error / uncertainty.

On the comment on diurnal dynamics, we consider that this is already clarified in our response to R1#7. See also our reply to R1#8 on the issue of uncertainty estimation.

R1#10b. Are error estimates and sources of errors given (and discussed in the article)? Unfortunately, no quantitative error estimates were given, although sources of errors are discussed in the article. Please see above.

This is now included in section 3.7 of the main text and in Appendix B (see the end of this document).

R1#11. Are common standards used for comparison? Unfortunately, I am not aware of any common standards for comparison against sapflow data. Lysimeter studies might be helpful, but I did not see any such studies cited in the manuscript.

We actually refer to lysimeters in the paper (L. 300), but we didn't add a specific citation there. The reason is that, while there are good examples of the use of lysimeters to compare against evapotranspiration (e.g. Pérez-Priego et al. 2017), we didn't find any study comparing lysimeters and transpiration under field conditions.

Regarding the standards for comparison, the best evidence available is the study by Flo et al 2019, which was conducted in the context of the SAPFLUXNET initiative and we cite in the paper. In that study, published sap flow calibrations (i.e. comparison of flows measured with sensors against flow measured using a standard, reference method) were compiled and synthesised.

Perez-Priego, O., El-Madany, T. S., Migliavacca, M., Kowalski, A. S., Jung, M., Carrara, A., Kolle, O., Martín, M. P., Pacheco-Labrador, J., Moreno, G. and Reichstein, M.: Evaluation of eddy covariance latent heat fluxes with independent lysimeter and sapflow estimates in a Mediterranean savannah ecosystem, Agricultural and Forest Meteorology, 236, 87–99, https://doi.org/10.1016/j.agrformet.2017.01.009, 2017.

R1#12. Are there any inconsistencies, implausible assertions or noticeable problems that would suggest the data are erroneous (or worse)? Most datasets included in the database have already been used and published elsewhere but I did not assess whether the results of the original publications can be reproduced using the database, nor did I see such an analysis as part of the data quality checks.

See our response to the comment R1#6.

R1#13. Given that most of the analysis is done with R, it would have been great if the authors included code to reproduce the figures and tables presented in this manuscript, and, perhaps for future data submissions, to reproduce figures in the original publications related to each dataset. Are figures and tables correct and of high quality? Since the code used to generate the figures and tables in the manuscript was not provided, I was not able to verify within the time I had for this review.

The code to reproduce the figures in this paper is available in the following Github repository: https://github.com/sapfluxnet/sfn_datapaper.

R1#14. Is the data set publication, as submitted, of high quality? Yes. Finally: By reading the article and downloading the data set, would you be able to understand and (re-)use the data set in the future? Very likely.

Thanks again for your positive comments.

DETAILED COMMENTS

L290: "understanding of"

Thanks, changed.

L295: This sentence does not make much sense to me: "In practice, transpiration is relatively easy to isolate from the bulk evaporative flux, evapotranspiration, only from the leaf to the plant levels." First of all, it is not easy to separate transpiration from ET at the leaf level, as the only other component of ET at leaf level is evaporation of intercepted water, which is not easily separated from transpiration. Secondly, transpiration measurement at the plant level is exactly what sapflow is about, so why are you saying that it is relatively easy?

In this sentence we refer to measurements at the leaf (using gas exchange cuvettes) and the plant level (using whole-plant cuvettes or gravimetric measurements). In both cases, soil evaporation can be neglected or minimised, by covering (or excluding) soil surfaces and thus preventing evaporation. We have now rewritten this part of the text to make it clear:

Conceptually, transpiration can be quantified at different organisational scales: leaves, branches and whole plants, ecosystems and watersheds. In practice, transpiration is relatively easy to isolate from the bulk evaporative flux, evapotranspiration, when measuring in a dry canopy, at the leaf or the plant level. However, in terrestrial ecosystems, evapotranspiration includes evaporation from the soil and from water-covered surfaces, including plants.

L314: When stating that sap flow sensors can be deployed in almost any terrestrial ecosystem, it would be important to point out that they are limited to woody plants (unless I am wrong).

Sap flow can also be measured in herbaceous species, or other species with non-woody stems for example see our reference to the work by Baker and Van Bavel 1987, already cited in the text and the new citation of the study by Lu et al. (2004) measuring sap flow in banana plants. This is the new text:

Whole-plant measurements of water use obtained with thermometric sap flow methods provide estimates of water flow through plants from sub-daily to interannual timescales, and have been mostly applied in woody plants, although several studies have measured sap flow on herbaceous species (Baker and Van Bavel, 1987; Skelton et al. 2013) and non-woody stems (e.g. Lu et al. 2004).

Baker, J. M. and Van Bavel, C. H. M.: Measurement of mass flow of water in the stems of herbaceous plants, Plant, Cell & Environment, 10(9), 777–782, https://doi.org/10.1111/1365-3040.ep11604765, 1987. Lu, P., Woo, K. C. and Liu, Z. T.: Estimation of whole-transpiration of bananas using sap flow measurements, Journal of Experimental Botany, 53(375), 1771–1779, https://doi.org/10.1093/jxb/erf019, 2002.

L338: Where are the quality assurance and control procedures described?

This is summarised in Fig. 1, explained in sections 2.2-2.5 (including Tables S1 and S2) and with all details in the sapfluxnetQC1 package.

L364: What do you mean by "methodological uncertainty"? Uncertainty about what method was used for a given dataset due to inadequate documentation, or uncertainty about what is the most suitable method for a given case? Could you clarify in the text?

The paragraph refers to sap flow methods in general, not specifically to SAPFLUXNET. Methodological details associated to each sap flow time series are well-documented in the plant metadata, and we have included a dedicated methodological section when we describe the database (section 3.2) and a new text on uncertainty quantification (section 3.7).

We have modified the last sentence of the paragraph to make clear that we refer to methodological variability arising from differences between and within methods. Here's the text:

Apart from these different methodologies, within each sap flow method sensor design (Davis et al. 2012) and data processing (Peters et al. 2018) can vary, resulting in relatively high levels of methodological variability comparable to those in other areas of plant ecophysiology.

L372: This sounds as if the stem heat balance method was superior to the others. Could you clarify?

Yes, we meant that, in contrast to most methods using linear heaters and thus heating

a small region of the sapwood, heat balance methods typically measure entire stems or large sections of sapwood and are more spatially representative. This has been clarified in the text:

Except for stem heat balance methods, which typically measure entire stems or large sapwood sections, most sap flow measurements need to be spatially integrated to account for radial (Berdanier et al., 2016; Cohen et al., 2008; Nadezhdina et al., 2002; Phillips et al., 1996) and azimuthal (Cohen et al., 2008; Lu et al., 2000; Oren et al., 1999a) variation of sap flow within the stem to obtain an estimate of whole-plant water use (Čermák et al., 2004).

L432-434: What does the Hampel filter do? How does it identify outliers and estimate correct values? Where is the code used to determine out of range values and outliers? Where is this R Shiny application? I could not find it at https://github.com/sapfluxnet. What is the role of expert knowledge here? Why were out of range values replaced by NA and outliers by the Hampel filter value?

The Hampel filter calculates a rolling median within a given temporal window and estimated data deviations from this value. If a given data point lies a number of standard deviations from the median, the data point is flagged as an outlier. We used a conservative setup for the number of standard deviations to really detect instrumental problems (10 SD). The code is the sapfluxnetQC1 package, and more specifically in the following function: https://github.com/sapfluxnet/sapfluxnetQC1/blob/c66115798d0c0814f399b1df6505f146594b6bdf/R/qc_outliers_and_rang

The Shiny application is also coded in the sapfluxnetQC1 package (we only show a screenshot in Fig. S3): https://github.com/sapfluxnet/sapfluxnetQC1/blob/c66115798d0c0814f399b1df6505f146594b6bdf/R/outliers_app.R

All automatic detection procedures (out of range checks, outlier checks) only flagged potential errors in the data, which had to visually confirmed before data replacements or deletions. Here's where the expert judgement is needed to decide whether

flagged data points were actually errors. This was needed in this very heterogeneous database, as a uniform, fully-automatic procedure to quality control time series with very different characteristics (e.g. periodic behaviour for radiation, random variation for precipitation or wind speed, highly variable and diverse diel sap flow patterns) is difficult to implement.

As for the reason to replace out of range values by NA and outliers by the Hampel filter value, it was mainly because out of range values were first removed to prevent potentially erroneous values affecting the calculations of the Hampel filter.

L440: I shink the first "Granda et al." should be removed.

Done, thanks.

L445: Was sap flow per unit leaf area computed even for deciduous species using constant leaf area all year round? Does it make sense to do this?

We are fully aware that we're not accounting for seasonal variation in leaf area when reporting leaf-area related fluxes but the environmental metadata associated to each dataset contains information on whether species or stand level variations in leaf area are available from the contributors. We have clarified this in the text:

If leaf area was supplied, we also calculated sap flow per unit leaf area, but note that this transformation does not take into account the seasonal variation in leaf area; we document in the metadata for which datasets this information could be available from data contributors.

L450-465: These details could be moved to the appendix, as I did not find it very informative without seeing the database.

The entire section 2 describes the full data workflow and we think that this should culminate with the description of the actual data structure, and specifically one of the main benefits which is having each dataset encapsulated in a consistent data structure (sfn_data objects). Another reason to place this explanation here is that we report how

dataset codes are formed, which is something that is then needed to understand the description of the database contents in section 3.

L490: Grammar: A sentence should not start with "Because".

We think that the use of 'Because' at the beginning of the sentence is not incorrect, as long as there is a clause afterwards completing the sentence. See for example, https://www.scribbr.com/academic-writing/myth-its-incorrect-to-start-a-sentence-with-because/

L503: By how much did HD and CHD underestimate sap flow rates?

We have added this to the text:

This lack of calibrations may be relevant for the more empirical heat dissipation methods (HD and CHD), which have been shown to consistently underestimate sap flow rates by 40% on average (Flo et al., 2019; Peters et al., 2018; Steppe et al., 2010).

L505: Grammar: reads as if plants were doing the data processing.

Thanks for spotting this, changed:

For a large number of plants measured with the HD method, and all plants measured with the HPTM method, there was not any radial integration procedure reported

L510: What does it mean for the upscaling? Could it not be done at all for datasets with missing sapwood area or depth?

This is a general statement on the overall availability of scalars which may be needed to interconvert sap flow between plant and sapwood area basis. The implications for upscaling depend on the dataset and the goal; for instance, if one dataset already reported sap flow per plant, the fact that sapwood area is missing is less relevant if one wants to upscale to the stand level using the metadata provided in sapfluxnet: stand basal area and the percentage basal area for each species.

[Figure]

L529: Is there any reliable way to estimate sapwood area for trees where this information was not provided?

In the new appendix we provide an example where sapwood areas were estimated using a site-specific allometry and we show how the uncertainty derived from sapwood estimation is lower than methodological uncertainty. We could address using the raw data from the allometric relationship, but these data are not available in SAPFLUXNET; we in fact suggest that adding this information to SAPFLUXNET could be useful to better document the upscaling process; here's the text:

The impact of not accounting for radial and circumferential variability when scaling single-point measurements of sap flow to the whole-plant level can be important (Merlin et al., 2020), but the estimation of sapwood area can also cause large errors if it is not accurately determined (Looker et al., 2016). SAPFLUXNET does not provide information on the method employed to quantify sapwood area (e.g. visual estimation with or without the application of dyes, indirect estimation through allometries at species or site levels) or on the accuracy of sapwood area data. This precludes uncertainty estimation at the individual level (Appendix Fig. B3). Future developments in the SAPFLUXNET data structure could include this information as metadata to better document the sensor-to-plant scaling process.

Allometric relationships could be obtained from the plant metadata in SAPFLUXNET (e.g. sapwood area as a function of tree basal area), at the species level or higher taxonomic levels (e.g. Angiosperms, Gymnosperms). Lastly, if species-level allometries cannot be obtained from SAPFLUXNET, external sources could be examined (e.g. BAAD).

Falster, D. S., et al: BAAD: a Biomass And Allometry Database for woody plants, Ecology, 96(5), 445–1446, 2015.

L. 529b. I did not understand what role tree height plays for the use and/or interpretation of the data.

SAPFLUXNET has been designed with a focus on the whole-plant level and tree height is a relevant variable influencing water transport in plants. We have made this clearer in section 4.1: by citing the recent highly relevant paper by Liu et al (2019); this is the text:

The availability of global sap flow data at sub-daily time resolution and spanning entire growing seasons will allow focusing on how maximum water use and its environmental sensitivity varies with plant-level attributes such as stem diameter (Dierick and Hölscher, 2009; Meinzer et al., 2005), tree height (Novick et al., 2009; Schäfer et al., 2000), hydraulic (Manzoni et al., 2013; Poyatos et al., 2007) and other plant traits (Grossiord et al., 2019; Kallarackal et al., 2013). SAPFLUXNET thus provides an unprecedented tool to understand how structural and physiological traits coordinate with each other (Liu et al. 2019), how these traits translate to whole-plant regulation of water fluxes (McCulloh et al., 2019), and how this integration determines drought responses (Choat et al., 2018) and post-drought recovery patterns (Yin and Bauerle, 2017).

Liu, H., Gleason, S. M., Hao, G., Hua, L., He, P., Goldstein, G. and Ye, Q.: Hydraulic traits are coordinated with maximum plant height at the global scale, Science Advances, 5(2), eaav1332, https://doi.org/10.1126/sciadv.aav1332, 2019.

L583: What does the "compared to..." mean? PAR and Rnet were provided less often than shortwave?

Yes, we have now clarified this in the text:

For radiation variables, shortwave radiation was most often provided, compared to photosynthetically active and net radiation, which were less provided; only 8 out of 202 datasets do not have any accompanying radiation data.

L. 583b. Was shortwave converted to PAR or vice versa to obtain a homogenous dataset?

Yes, when direct interconversions between variables were possible, these were applied

(Table S2).

L594: It would be good to mention that sub-daily sap flow variation is not necessarily consistent with sub-daily variation in transpiration.

This (L. 594-598) is a general statement on the database to highlight its potential to address patterns at different timescales (from sub-daily to growing season). Please note that we don't refer to 'transpiration' here but to 'water use', and maximum rates of plant water transport are also interesting to analyse, in terms of their environmental sensitivity and relationship with plant attributes.

We fully agree with the reviewer that sap flow is not transpiration at short timescales, and this has now been further clarified in the text; see our response to comment R1#7.

L599: What do you mean by "scale up to"? Maybe "translate to"?

We have changed this:

SAPFLUXNET thus provides an unprecedented tool to understand how structural and physiological traits coordinate with each other (Liu et al. 2019), how these traits translate to whole-plant regulation of water fluxes (McCulloh et al., 2019), and how this integration determines drought responses (Choat et al., 2018) and post-drought recovery patterns (Yin and Bauerle, 2017).

L504: How could the data be used to estimate hydraulic conductivity?

We believe that the reviewer refers to L.604. Sap flow data has been used in combination with water potential gradients from soil to leaves to estimate whole-plant hydraulic conductance. We have clarified the text and cited the work by Cochard et al 1996 showing this estimation:

If combined with water potential measurements, sap flow data can be used to estimate whole-plant hydraulic conductance and study its response to drought (e.g., Cochard et al., 1996), as well as the recovery of the plant hydraulic system after drought.
Cochard, H., Bréda, N. and Granier, A.: Whole tree hydraulic conductance and water loss regulation in Quercus during drought: evidence for stomatal control of embolism?, Annales des Sciences Forestières, 53, 197–206, 1996.

L605-610: This may be misleading, as diurnal patterns in sap flow do not necessarily translate to diurnal patterns in transpiration, due to capacitance. So the relevant time lags are not only between evaporative demand and sap flow, as stated here, but also between sap flow and transpiration. This needs to be clarified to avoid confusion.

We agree with the reviewer, and we have modified the sentence accordingly:

Hysteresis in diel sap flow relationships with evaporative demand and time-lags between transpiration and sap flow, are two linked phenomena likely arising from plant capacitance and other mechanisms (O'Brien et al., 2004; Schulze et al., 1985), that also influence diel evapotranspiration dynamics (Matheny et al., 2014; Zhang et al., 2014).

L616: Why would SAPFLUXNET allow quantification of nocturnal sap flow for data sets less suitable for quantification of night-time fluxes? More explanation would be helpful here.

It's true that our phrasing was a bit unclear. What we meant is that metadata documents which datasets may be useful to study night-time fluxes. We propose the following sentence:

SAPFLUXNET includes metadata to identify methods (e.g. HRM; Burgess et al. 2001) and data processing approaches (zero-flow determination method in 'pl_sens_cor_zero', Appendix Table A5) that can help identify suitable datasets to quantify night-time fluxes.

L625: How are treatments documented in the database? Qualitatively or quantitatively

The documentation of the treatments has not been standardised and they are documented in a qualitative way, although some quantitative information can be found in the

corresponding treatment description in the metadata ('st_treatment' or 'pl_treatment'). This has been clarified in the text:

The SAPFLUXNET database includes datasets with experimental manipulations, applied either at the stand or at the individual level, qualitatively documented in the metadata (Table 3).

L658: Has any of this been attempted already, and if so, how big are the uncertanties?

We report a preliminary application of this in Poyatos et al 2020b, but we didn't quantify the uncertainties there. In the revised version we provide elements to quantify uncertainty caused by different sources impacting plant-level sap flow (see the response to comment R1#8). Propagating these uncertainties to the stand level would need to introduce information or assumptions on the sampling of the trees measured in each dataset. We consider this an interesting avenue to pursue when developing ecosystem upscaling procedures, but this is currently out of the scope of this plant-level database.

Poyatos, R., Flo, V., Granda, V., Steppe, K., Mencuccini, M. and Martínez-Vilalta, J.: Using the SAPFLUXNET database to understand transpiration regulation of trees and forests, Acta Hortic., (1300), 179–186, https://doi.org/10.17660/ActaHortic.2020.1300.23, 2020.

We also note that the availability of global sap flow data in SAPFLUXNET will precisely stimulate the development of new frameworks to upscale uncertainties, as occurred with the compilation of global eddy flux data in the FLUXNET network. Here is the new text in section 5.1:

Overall, this first global compilation of sap flow data will allow addressing uncertainties in sap flow upscaling in space and time in the same way that the development of FLUXNET stimulated the quantification and aggregation of uncertainties for eddy flux data (Richardson et al. 2012).

L676: To know if this up-scaling is promising, it would be important to know the uncertainty in sap flow derived individual-based or sapwood area based transpiration.

Please see our reply to our previous comment and to comment R1#8.

L704: How could uncertainty propagation ever be done for the commercial methods? This might be a good place to emphasise the need for open source data-processing software.

We agree with the reviewer, and we have added a sentence showing how of open software can contribute to obtain more robust estimates of sap flow.

Open source software also allows a seamless integration of different data processing approaches and the implementation of species-specific calibrations, which can contribute to obtain more robust estimations of sap flow and facilitate replicability (Peters et al. 2021).

Peters, R. L., Pappas, C., Hurley, A. G., Poyatos, R., Flo, V., Zweifel, R., Goossens, W. and Steppe, K.: Assimilate, process and analyse thermal dissipation sap flow data using the TREX r package, Methods in Ecology and Evolution, https://doi.org/10.1111/2041-210X.13524, 2021.

L710-711: How accurate, how big is the under-estimation?

We found that a 40% underestimation, on average, when using the original calibration. This has been now clarified in the text, specifically in section 3.7:

The analysis of calibration data also showed that HD, the most represented method by far, underestimates water flow, on average, by 40% (Flo et al., 2019) when using the original calibration (Granier et al., 1985; 1987).

L711: Grammar: don't start a sentence with "Because"

We think that the use of 'Because' at the beginning of the sentence is not incorrect, as long as there is a clause afterwards completing the sentence. See for example, https://www.scribbr.com/academic-writing/myth-its-incorrect-to-starta-sentence-with-because/

L715: So what error bands should we expect for the data before and after corrections?

This is now explained in section 3.7 and shown in the Appendix (see the end of this document). Related to the uncertainties in data processing that we mention in this section 5.1. we have clarified that assuming zero flow at night typically leads to sap flow underestimation; this is the text:

The determination of zero flow conditions (baselining) can also have significant impacts on the quantification of absolute flow for several methods (Smith and Allen, 1996; Steppe et al., 2010), with commonly applied assumptions of pre-dawn zero flow typically leading to underestimation of sap flow (Peters et al. 2018). The different baselining approaches are also documented in the metadata to inform data syntheses and/or to selectively apply correction factors.

L728: How large can the errors due to sapwood area estimation errors be?

In this revised version, we have compared the methodological uncertainty associated with the HD method with that derived from the estimation of sapwood using a site-specific allometric relationship for a specific dataset for which we have the raw data used to obtain such relationship (ESP_VAL_SOR). As Fig. B3 shows, the contribution of methodological uncertainty is much larger than that by sapwood estimation.

We note that this comparison cannot be done for all SAPFLUXNET datasets because this information is not available. This is why, at a minimum, future iterations of SAPFLUXNET should probably contain some metadata as to how was sapwood estimated in the sampled trees (L. 730-731).

L773: Could you add the relevant packages used for data processing and quality control, some of which are mentioned in https://github.com/sapfluxnet, and specivy which version of each package was used?

We think that this information belongs to section 2 where the data workflow is

described. In our opinion, this section 6 could also be placed at the end of section 2, but the structure of ESSD mandates that the paper ends with a 'Data availability' section before the 'Conclusions' Appendix B: Uncertainty estimation in sap flow measurements in the SAPFLUXNET database Here we will show examples of uncertainty estimation for sap flow data in the SAPLUXNET database. We will address three main sources of uncertainty which affect plant-level estimates of sap flow: (i) methodological uncertainty, (ii) sapwood area uncertainty and (iii) radial integration uncertainty. Methodological uncertainty was estimated using the data in the global meta-analysis of sap flow calibrations by Flo et al. (2019) as published in Flo et al. (2021). This estimation can be applied for the main sap flow density methods. We predicted the standard error (SE) of sub-daily sap flow density by fitting, for each method, linear mixed models of reference flow (i.e. using a gravimetric method or others employed as reference standards in calibration studies) as a function of measured flow, including the individual calibration as a random intercept factor (Table B1, Figure B1). This model shows that HPTM presents the highest uncertainty and that this method and CHP are the ones showing larger uncertainties at low flows, while HD and CHD show lower relative uncertainty at high sap flow density (Figure B1, B2). We also show in Figure B3a the effect of applying the bias correction factor for uncalibrated heat dissipation probes obtained from the meta-analysis by Flo et al. (2019). Uncertainty in the determination of sapwood area can arise when allometric relationships are used to estimate sapwood area, because this area is then applied to upscale sap flow density values to the whole-plant. This uncertainty can be accounted for if the original data employed to obtain the allometry are available. Using these data for one of the datasets in SAPFLUXNET (ESP_VAL_SOR), we first predicted sapwood area, together with the upper and lower bounds of its 68% predictive interval (equivalent to one SE). Then, we estimated the corresponding mean sap flow and its 68% uncertainty interval (Figure B3a). In this case, methodological uncertainty was larger than that caused by sapwood area estimation (Figure B3b). Total combined uncertainty (i.e. methodological and sapwood) was obtained by adding their squared

values and then taking the square root, following error propagation theory (Figure B3c). In this example tree, total uncertainty for instantaneous values is around 400-500 cm3 h-1, resulting in a high uncertainty for low flows but low relative uncertainty for higher flows, reaching 13% at peak flows on the 6th of June (Figure B3c). When expressed as daily means, this uncertainty will be reduced as temporal averaging decreases the uncertainty by a factor equal to the inverse of the root square of the number of observations within a day (Richardson et al. 2012). In the same example (Figure B3c), A day with high daily mean flow will also show lower relative uncertainty (June 6th ,1589 $\pm$ 45 cm3 h-1, 3%) compared to one with lower daily mean flow (May 30th, 237 $\pm$ 45 cm3 h-1, 19%). Finally, when no information on the variation of sap flow along the sapwood is available, radial integration of point measurements of sap flow density and associated uncertainty can be obtained by applying generic radial profiles according to wood porosity (Berdanier et al. 2016) as implemented in the R package 'sapflux' (https://github.com/berdaniera/sapflux). An example application of this procedure shows how different uncertainty bounds can be obtained depending on wood anatomy (Figure B4). In addition, this application shows how assuming an uniform radial profile for ring-porous or diffuse-porous can lead to substantial underestimation of whole-plant sap flow, compared to a lower impact for tracheid-bearing species.

Please also note the supplement to this comment:
https://essd.copernicus.org/preprints/essd-2020-227/essd-2020-227-AC1-supplement.pdf

———————————————————

**Fig. 1.** Figure B1. Methodological uncertainty estimation in sap flow density measurements based on the data from the global metanalysis of sap flow calibrations in Flo et al. (2019).

**a) Heat dissipation**

**b) Compensation heat pulse**

**c) Heat ratio**

**Fig. 2.** Figure B2. Sub-daily time series of sap flow and methodological uncertainty estimations (one standard error) according to the model in Fig. B1.

**Fig. 3.** Figure B3. An example of sap flow uncertainty estimation and bias correction for a Pinus sylvestris tree (ESP_VAL_SOR_Js_Ps_12) measured using heat dissipation sensors.

[Figure]

**Sap flow** — **Sap flow radially integrated**

(a) Quercus rubra,Ring–porous — USA_UMB_CON_Qru_Js_10

(b) Pinus strobus,Tracheid — USA_UMB_CON_Pst_Js_2

(c) Populus grandidentata,Diffuse–porous — USA_UMB_CON_Pgr_Js_3

Sap flow, $cm^3 h^{-1}$

Jun 05  Jun 07  Jun 09  Jun 11  Jun 13  Jun 15

**Fig. 4.** Figure B4. Effects of a generic radial integration on sap flow data originally supplied without any radial integration procedure.

| Method | Intercept | Slope |
|--------|-----------|-------|
| HD | 1.49 | 0.01 |
| CHP | 2.65 | 0.03 |
| HR | 0.76 | 0.03 |
| HPTM | 7.75 | 0.04 |
| CHD | 2.03 | 0.01 |
| HFD | 1.05 | 0.01 |

**Fig. 5.** Table B1.Table B1. Fixed-effects coefficients from the linear mixed models fitting reference sap flow density as a function of measured sap flow density global sap flow calibrations data.

---

## Author Comment (AC2) · 3 Mar 2021

Please see the supplement file for a formatted version of this document (comments and replies in different font and style).

General comments: The SAPFLUXNET database represents an excellent undertaking by Poyatos et al., and these authors should be congratulated for a fine achievement. The paper is well laid out and clear and I only have a few minor comments for the authors that I hope will improve the manuscript. It will be good to see this paper/dataset published and I look forward to being able to contribute to it and to make use of it in the future.

We would like to thank the reviewer for his positive views and his appreciation of our

work.

One suggestion for an improvement is to highlight an example of where the dataset has been used to answer an experimental question. For example, is there a dataset that can demonstrate the impact of an abiotic stress on sap flow for different species?

We understand that by 'the dataset' the reviewer is referring to the entire database. We have a manuscript under revision and several other manuscripts in the pipeline that use the database to address questions regarding the environmental controls on tree water use. Preliminary contributions using different subsets of sapfluxnet have been cited in the manuscript:

De Cáceres, M., Mencuccini, M., Martin-StPaul, N., Limousin, J.-M., Coll, L., Poyatos, R., Cabon, A., Granda, V., Forner, A., Valladares, F. and Martínez-Vilalta, J.: Unravelling the effect of species mixing on water use and drought stress in Mediterranean forests: A modelling approach, Agricultural and Forest Meteorology, 296, 108233, https://doi.org/10.1016/j.agrformet.2020.108233, 2021.

Nelson, J. A., Pérez‐Priego, O., Zhou, S., Poyatos, R., Zhang, Y., Blanken, P. D., Gimeno, T. E., Wohlfahrt, G., Desai, A. R., Gioli, B., Limousin, J.-M., Bonal, D., Paul‐Limoges, E., Scott, R. L., Varlagin, A., Fuchs, K., Montagnani, L., Wolf, S., Delpierre, N., Berveiller, D., Gharun, M., Marchesini, L. B., Gianelle, D., Šigut, L., Mammarella, I., Siebicke, L., Black, T. A., Knohl, A., Hörtnagl, L., Magliulo, V., Besnard, S., Weber, U., Carvalhais, N., Migliavacca, M., Reichstein, M. and Jung, M.: Ecosystem transpiration and evaporation: Insights from three water flux partitioning methods across FLUXNET sites, Global Change Biology, 26(12), 6916–6930, https://doi.org/10.1111/gcb.15314, 2020.

Poyatos, R., Flo, V., Granda, V., Steppe, K., Mencuccini, M. and Martínez-Vilalta, J.: Using the SAPFLUXNET database to understand transpiration regulation of trees and forests, Acta Hortic., (1300), 179–186, https://doi.org/10.17660/ActaHortic.2020.1300.23, 2020.

Specific comments:

Line 290: The main line of inquiry is not how plants regulate transpiration, but rather how transpiration varies with abiotic factors and along environmental gradients in different species. Perhaps the wording could be changed to something like "An improved understanding of transpiration and how variable this process is under different abiotic conditions, along environmental gradients and in different species is thus needed to. . ."

Thanks for the suggestion. We have included the reviewer's idea in the following sentence:

An improved understanding of transpiration and its regulation along environmental gradients and across species is thus needed to predict future trajectories of land evaporative fluxes and vegetation functioning under increased drought conditions driven by global change.

Line 312: Several studies have also quantified sap flow of graminoid species; see for example Skelton et al. 2012 (There are other papers).

Thanks for the suggestion, it's true that there are more recent examples following Baker and Van Bavel (1987). We have now cited Skelton et al. 2013.

Whole-plant measurements of water use using thermometric sap flow methods provide estimates of water flow through plants from sub-daily to interannual timescales, and have been mostly applied in woody plants, although several studies have measured sap flow on herbaceous species (Baker and Van Bavel, 1987; Skelton et al. 2013).

Line 313: See Clearwater et al. 2009 for a non-invasive approach using non-invasive external probes. Also cite Clearwater et al. 2009 on line 351.

Thanks for the suggestion. We have placed the suggested reference along with Sakuratani's pioneering work on external stem heat balance measurements and Helfter et al. (2007) study on a laser-based heating system, in the text where we comment on

the differences between internally- and externally-heated systems.

Both heating and temperature sensing can be done either internally, by inserting needle-like probes containing electrical resistors (or electrodes for some methods) and temperature sensors into the sapwood (Vandegehuchte and Steppe, 2013), or externally; these latter systems being especially designed for small stems and non-lignified tissues (Clearwater et al. 2009; Helfter et al. 2007; Sakuratani 1981).

References 1. Skelton, R. P., West, A. G., Dawson, T. E. & Leonard, J. M. External heat-pulse method allows comparative sapflow measurements in diverse functional types in a Mediterranean-type shrubland in South Africa. Funct. Plant Biol. 40, 1076–1087 (2013). 2. Clearwater, M. J., Luo, Z., Mazzeo, M. & Dichio, B. An external heat pulse method for measurement of sap flow through fruit pedicels, leaf petioles and other small-diameter stems. Plant. Cell Environ. 32, 1652–1663 (2009).

Line 361: It would be best to avoid acronyms to make the sentence (and other sentences) easier to read. It is much easier to read "heat dissipation or compensation heat pulse" in this sentence than to keep referring to where the acronym was first mentioned.

We have replaced acronyms by the full name of each method in this sentence:

The suitability of a certain method in a given application largely depends on plant size and the flow range of interest (Flo et al., 2019), but heat dissipation and compensation heat pulse are the most widely used (Flo et al., 2019; Poyatos et al., 2016).

We have reviewed other parts of the text with a heavy use of acronyms (e.g. section 3.2), but we think that replacing acronyms by complete names would sometimes lead to unnecessarily long sentences and would hinder readability (see L.504-507).

Line 405: Are there plans for this site to be maintained? Can people continue to contribute? Perhaps state this up front.

We would like to keep SAPFLUXNET updated and receive more datasets, but this demands having some staff available to perform all the data ingestion process and to

solve any arising issues with the help of data contributors. We will definitely search ways to make this possible in the near future.

In section 2.2 we state that the data contribution period was open in 2016-2017 (L.392) and we mention the possibility of opening new data contribution periods in section 5 (L. 759-761).

Please also note the supplement to this comment:
https://essd.copernicus.org/preprints/essd-2020-227/essd-2020-227-AC2-supplement.pdf

---

## Author Response (AR2)

CREAF. Campus UAB. Edifici C
08193 Cerdanyola del Vallès
(Barcelona)
Tel. + 34 93 581 13 12
Fax + 34 93 581 41 51
contacte@creaf.uab.cat
**www.creaf.cat |
blog.creaf.cat**

Dear Editor,

In response to the last comments by the referee, I have attached the files (final version and track changes version) with the changes requested. Briefly, these changes consist of:

- The publication of the code used to reproduce figures, supporting information and appendices in a Zenodo repository (https://doi.org/10.5281/zenodo.4727825). I have updated section 7 of the manuscript accordingly.
- Replacement of the DOI handle pointing to the SAPFLUXNET database version 0.1.5 in Zenodo (https://doi.org/10.5281/zenodo.3971689). This has been changed in the README file of the code repository (https://github.com/sapfluxnet/sfn_datapaper).

I would like to thank you for handling the manuscript and the reviewers for providing very useful feedback to improve the paper.

Best,

Rafael Poyatos,
on behalf of the co-authors

Barcelona, 29th April 2021